# CERTIFIABLY ROBUST POLICY LEARNING AGAINST ADVERSARIAL MULTI-AGENT COMMUNICATION

**Yanchao Sun**[†*]  **Ruijie Zheng**[†]  **Parisa Hassanzadeh**[‡]  **Yongyuan Liang**[§]
**Soheil Feizi**[†]  **Sumitra Ganesh**[‡]  **Furong Huang**[†]
[†] University of Maryland, College Park  {ycs, rzheng12, sfeizi, furongh}@umd.edu
[‡] JPMorgan AI Research  {parisa.hassanzadeh, sumitra.ganesh}@jpmchase.com
[§] Shanghai AI Lab  cheryllLiang@outlook.com

## ABSTRACT

Communication is important in many multi-agent reinforcement learning (MARL) problems for agents to share information and make good decisions. However, when deploying trained communicative agents in a real-world application where noise and potential attackers exist, the safety of communication-based policies becomes a severe issue that is underexplored. Specifically, if communication messages are manipulated by malicious attackers, agents relying on untrustworthy communication may take unsafe actions that lead to catastrophic consequences. Therefore, it is crucial to ensure that agents will not be misled by corrupted communication, while still benefiting from benign communication. In this work, we consider an environment with $N$ agents, where the attacker may arbitrarily change the communication from any $C < \frac{N-1}{2}$ agents to a victim agent. For this strong threat model, we propose a certifiable defense by constructing a message-ensemble policy that aggregates multiple randomly ablated message sets. Theoretical analysis shows that this message-ensemble policy can utilize benign communication while being certifiably robust to adversarial communication, regardless of the attacking algorithm. Experiments in multiple environments verify that our defense significantly improves the robustness of trained policies against various types of attacks.

## 1  INTRODUCTION

Neural network-based multi-agent reinforcement learning (MARL) has achieved significant advances in many real-world applications, such as autonomous driving (Shalev-Shwartz et al., 2016; Sallab et al., 2017). In a multi-agent system, especially in a cooperative game, communication usually plays an important role. By feeding communication messages as additional inputs to the policy network, each agent can obtain more information about the environment and other agents, and thus can learn a better policy (Foerster et al., 2016; Hausknecht, 2016; Sukhbaatar et al., 2016). However, such a communication-dependent policy may not make safe and robust decisions when communication messages are perturbed or corrupted. For example, suppose an agent is trained in a cooperative environment with benign communication, and it learns to trust all communication messages and utilize them. But during test time, there exists a malicious attacker perturbing some communication messages, such that this agent can be drastically misled by the false communication.

The robustness of policy against adversarial communication is crucial for the practical application of MARL. For example, when several drones execute pre-trained policies and exchange information via wireless communication, it is possible that messages get noisy in a hostile environment, or even some malicious attacker eavesdrops on their communication and intentionally perturbs some messages to a victim agent via cyber attacks. Moreover, even if the communication channel is protected by advanced encryption algorithms, an attacker may also hack some agents and alter the messages before they are sent out (e.g. hacking IoT devices that usually lack sufficient protection (Naik & Maral, 2017)). Figure 1 shows an example of communication attacks, where the agents are trained with benign communication, but attackers may perturb the communication during the test time. The attacker may lure a well-trained agent to a dangerous location through malicious message propagation and

---

*Part of this work was done when the first author was an intern at JPMorgan AI Research.

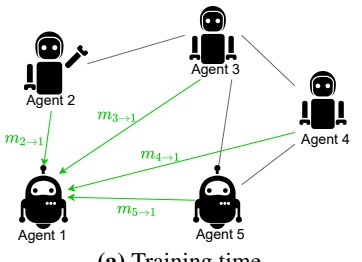 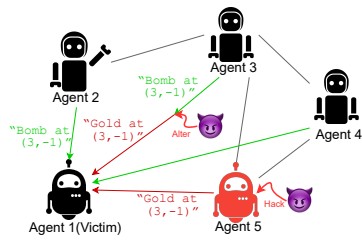

**(a)** Training time                  **(b)** Test time (in deployment)

**Figure 1:** An example of test-time communication attacks in a communicative MARL system. (a) During training, agents are trained collaboratively in a safe environment, such as a simulated environment. (b) In deployment, agents execute pre-trained policies in the real world, where malicious attackers may modify the benign (green) messages into adversarial (red) signals to mislead some victim agent(s).

cause fatal damage. Although our paper focuses on adversarial perturbations of the communication messages, it also includes unintentional perturbations, such as misinformation due to malfunctioning sensors or communication failures; these natural perturbations are no worse than adversarial attacks.

Achieving high performance in MARL through inter-agent communication while being robust to adversarial communication is a challenging problem due to the following reasons. **Challenge I**: Communication attacks can be stealthy and strong. The attacker may construct a false communication that is far from the original communication, but still semantically meaningful. In the example of Figure 1b, the attacker alters "Bomb" to "Gold", which can mislead the victim agent to the location of a bomb. But the victim, without seeing the groundtruth, cannot see the maliciousness from the message itself. Note that the widely-used $\ell_p$ threat model (Chakraborty et al., 2018) does not cover this situation. **Challenge II**: The attacker can even be *adaptive* to the victim agent and significantly reduce the victim's total reward. For instance, for a victim agent who moves according to the average of GPS coordinates sent by others, the attacker may learn to send extreme coordinates to influence the average. **Challenge III**: There can be more than one attacker (or an attacker can perturb more than one message at one step), such that they can collaborate to mislead a victim agent.

Although adversarial attacks and defenses have been extensively studied in supervised learning (Madry et al., 2018; Zhang et al., 2019) and reinforcement learning (Zhang et al., 2020b; Sun et al., 2022), there has been little discussion on the robustness issue against adversarial communication in MARL problems. Some recent works (Blumenkamp & Prorok, 2020; Xue et al., 2022; Mitchell et al., 2020) take the first step to investigate adversarial communications in MARL and propose several defending methods. However, these empirical defenses do not fully address the aforementioned challenges, and are not guaranteed to be robust, especially under adaptive attacks. In high-stakes applications, it is also important to ensure robustness with theoretical guarantees and interpretations.

In this paper, we address all aforementioned challenges by proposing a certifiable defense named **Ablated Message Ensemble (AME)**, that can guarantee the performance of agents when a fraction of communication messages get *arbitrarily* perturbed. Inspired by the ensemble methods which are proved to be the optimal defense against poisoning attacks under the iid sample setting (Wang et al., 2022), we propose to defend by ablation and ensemble of message sets, which tackles the challenging interactive decision-making under partially observable environments with correlated message samples. The main idea of AME is to make decisions based on multiple different subsets of communication messages. Specifically, for a list of messages coming from different agents, we train a *message-ablation policy* that takes in a subset of messages and outputs a *base action*. Then, we construct an *message-ensemble policy* by aggregating multiple base actions coming from multiple ablated message subsets. We show that when benign messages are able to reach some consensus, AME aggregates the wisdom of benign messages and thus is resistant to adversarial perturbations, no matter how strong the perturbation is. In other words, AME tolerates arbitrarily strong adversarial perturbations as long as the majority of agents are benign and uniting. Levine & Feizi (2020) use a similar randomized ablation idea to defend against $\ell_0$ attacks in image classification. However, they provide high-probability guarantee for classification, which is not suitable for sequential decision-making problems, as the guaranteed probability decreases when it propagates over timesteps.

Our contributions can be summarized as below:
**(1)** We formulate the problem of adversarial attacks and defenses in communicative MARL (CMARL).
**(2)** We propose a novel defense method, AME, that is certifiably robust against arbitrary perturbations

of up to $C < \frac{N-1}{2}$ communications under mild conditions, where $N$ is the number of agents.
**(3)** Experiment in four multi-agent environments shows that AME obtains significantly higher reward than baseline methods under both non-adaptive and adaptive attackers.

## 2 COMMUNICATIVE MUTI-AGENT REINFORCEMENT LEARNING (CMARL)

**Dec-POMDP Model.** We consider a Decentralised Partially Observable Markov Decision Process (Dec-POMDP) (Oliehoek, 2012; Oliehoek & Amato, 2015; Das et al., 2019) which is a multi-agent generalization of the single-agent POMDP models. A Dec-POMDP can be modeled as a tuple $\langle \mathcal{D}, \mathcal{S}, \mathcal{A}_\mathcal{D}, \mathcal{O}_\mathcal{D}, O, P, R, \gamma, \rho_0 \rangle$. $\mathcal{D} = \{1, \cdots, N\}$ is the set of $N$ agents. $\mathcal{S}$ is the underlying state space. $\mathcal{A}_\mathcal{D} = \times_{i \in \mathcal{D}} \mathcal{A}_i$ is the *joint* action space. $\mathcal{O}_\mathcal{D} = \times_{i \in \mathcal{D}} \mathcal{O}_i$ is the *joint* observation space, with $O$ being the observation emission function. $P : \mathcal{S} \times \mathcal{A}_1 \times \cdots \mathcal{A}_N \to \Delta(\mathcal{S})$ is the state transition function[1]. $R : \mathcal{S} \times \mathcal{A}_1 \times \cdots \mathcal{A}_N \to \mathbb{R}$ is the shared reward function. $\gamma$ is the shared discount factor, and $\rho_0$ is the initial state distribution.

**Communication Policy $\xi$ in Dec-POMDP.** Due to the partial observability, communication among agents is crucial for them to obtain high rewards. Consider a shared message space $\mathcal{M}$, where a message $m \in \mathcal{M}$ can be a scalar or a vector, e.g., signal of GPS coordinates. The communication policy of agent $i \in \mathcal{D}$, denoted as $\xi_i$, generates messages based on the agent's observation and interaction history. We use $m_{i \to j}$ to denote the message sent from agent $i$ to agent $j \neq i$. In practice, the communication policy can be hand-coded with domain knowledge, or learned with function approximators. Note that our paper focuses on how to defend against adversarial perturbations of existing communication, and *our defense is independent of the underlying communication policy $\xi_i$.*

**Acting Policy $\pi$ with Communication.** The goal of each agent $i \in \mathcal{D}$ is to maximize the discounted cumulative reward $\sum_{t=0}^{\infty} \gamma^t r^{(t)}$ by learning an acting policy $\pi_i$. When there exists communication, the policy input contains both its own interaction history, denoted by $\tau_i \in \Gamma_i$, and the communication messages $\mathbf{m}_{: \to i} := \{m_{j \to i} | 1 \leq j \leq N, j \neq i\}$. Similar to the communication policy $\xi$, we do not make any assumption on how the acting policy $\pi$ is formulated, as our defense mechanism introduced later can be plugged into any policy learning procedure.

## 3 PROBLEM SETUP: COMMUNICATION ATTACKS IN CMARL

Communication attacks in MARL has recently attracted increasing attention (Blumenkamp & Prorok, 2020; Tu et al., 2021; Xue et al., 2022) with different focuses, as summarized in Section 5. In this paper, we consider a practical and strong threat model where malicious attackers can arbitrarily perturb a subset of communication messages during test time.

**Formulation of Communication Attack: The Threat Model.** During test time, agents execute well-trained policies $\pi_1, \cdots, \pi_N$. As shown in Figure 1b, the attacker may perturb communication messages to mislead a specific victim agent. Without loss of generality, suppose $i \in \mathcal{D}$ is the victim agent. For notation simplicity, we assume agent $i$ receives messages from all other agents as is common in practice, resulting in $N - 1$ communication messages in total. But our defense also works for partially-connected communication graphs. We consider the sparse attack setup where up to $C$ messages can be arbitrarily perturbed at every step, among all $N - 1$ messages. Here $C$ is a reflection of the attacker's attacking power. The victim agent has no knowledge of which messages are adversarial. This type of attack is related to $\ell_0$ perturbations to the input (Levine & Feizi, 2020), as detailed in Appendix C. We make the following mild assumption for the attacking power.

**Assumption 3.1** (Attacking Power). *An attacker can **arbitrarily** manipulate fewer than a half of the communication messages, i.e., $C < \frac{N-1}{2}$.*

This is a realistic assumption, as it takes the attacker's resources to hack or control each communication channel. It is less likely that an attacker can change the majority of communications among agents without being detected. Note that this is a *very strong threat model* as the $C$ attacked messages can be arbitrarily perturbed based on the attacker's attacking algorithm. Compared to the commonly used $\ell_p$ attacks, the above threat model can work for broader applications, such as adding a patch to an image, replacing a word in a sentence, and etc.

We do not make any assumption on what attack algorithm the attacker uses, i.e., how a message is perturbed. That is, we aim to achieve generic and provable robustness under a wide range of practical

---

[1]$\Delta(\mathcal{X})$ denotes the space of probability distributions over space $\mathcal{X}$.

scenarios. In practice, we do not require the learner to know the exact $C$, and having an upper bound of $C$ suffices to obtain the guarantees we introduce in the next section.

## 4 PROVABLY ROBUST DEFENSE FOR CMARL

In this section, we present our defense algorithm, *Ablated Message Ensemble (AME)*, against test-time communication attacks in CMARL. We first introduce the algorithm design in Section 4.1, then present the theoretical analysis in Section 4.2. Section 4.3 discusses an extension of AME.

### 4.1 ABLATED MESSAGE ENSEMBLE (AME)

Our goal is to learn and execute a robust policy for any agent in the environment, so that the agent can perform well in both a non-adversarial environment and an adversarial environment. To ease the illustration, we focus on robustifying an arbitrary agent $i \in \mathcal{D}$, and the same defense is applicable to all other agents. We omit the agent subscript $_i$, and denote the agent's observation space, action space, and interaction history space as $\mathcal{O}$, $\mathcal{A}$, and $\Gamma$, respectively.

Let $\mathbf{m} \in \mathcal{M}^{N-1}$ denote a set of $N-1$ messages received by the agent. Then, we can build an ablated message subset of $\mathbf{m}$ with $k$ randomly selected messages, as defined below.

**Definition 4.1** (*$k$-Ablation Message Sample (k-Sample)*). *For a message set $\mathbf{m} \in \mathcal{M}^{N-1}$ and any integer $1 \leq k \leq N-1$, define a k-ablation message sample (or k-sample for short), $[\mathbf{m}]_k \in \mathcal{M}^k$, as a set of $k$ randomly sampled messages from $\mathbf{m}$. Let $\mathcal{H}_k(\mathbf{m})$ be the collection of all unique k-samples of $\mathbf{m}$, and thus $|\mathcal{H}_k(\mathbf{m})| = \binom{N-1}{k}$.*

We propose Ablated Message Ensemble (AME), a generic defense framework that can be fused with any policy learning algorithm. Motivated by the fact that the benign messages sent from other agents usually contain overlapping information of the environment that may suggest similar decisions, the **main idea** of AME is to make decisions based on the *consensus* of the benign messages. AME has two phases: the training phase where agents are trained in a clean environment, and a testing/defending phase where communications may be perturbed.

**Training Phase with Message-Ablation Policy $\hat{\pi}$ (Algorithm 1).** During training time, the agent learns a *message-ablation policy* $\hat{\pi} : \Gamma \times \mathcal{M}^k \to \mathcal{A}$ which maps its own interaction history $\tau$ and a random k-sample $[\mathbf{m}]_k \sim \mathrm{Uniform}(\mathcal{H}_k(\mathbf{m}))$ to an action, where $\mathrm{Uniform}(\mathcal{H}_k(\mathbf{m}))$ is the uniform distribution over all k-samples from the message set $\mathbf{m}$ it receives. Here $k$ is a user-specified hyper-parameter selected to satisfy certain conditions, as discussed in Section 4.2. The training objective is to maximize the cumulative reward of $\hat{\pi}$ based on randomly sampled k-samples in a non-adversarial environment. Any policy optimization algorithm can be used for training. Although by ablating a subset of messages, the message-ablation policy may compromise some natural reward (which is typical for robust training methods), it can significantly enhance the robustness of agents when attacks exist, with our design of test-time defense mechanism introduced below.

**Defending Phase with Message-Ensemble Policy $\widetilde{\pi}$ (Algorithm 2).** Once we obtain a reasonable message-ablation policy $\hat{\pi}$, we can execute it

---

**Algorithm 1** Training Phase of AME

1: **Input:** ablation size $k$
2: Initialize $\hat{\pi}_i$ for every agent $i \in [N]$.
3: **repeat**
4:     **for** $i = 1$ **to** $N$ **do**
5:         Receive $\mathbf{m}_{:\to i}$, $o_i$ and update history $\tau_i$
6:         Randomly sample $k$ messages and form a k-sample $[\mathbf{m}_{:\to i}]_k$
7:         Take action $a_i \leftarrow \hat{\pi}_i(\tau_i, [\mathbf{m}_{:\to i}]_k)$
8:         Update policy $\hat{\pi}_i$
9:     **end for**
10: **until** end of training
11: **Output:** $\hat{\pi}_i, \forall i \in [N]$

---

**Algorithm 2** Defending Phase of AME

1: **Input:** ablation size $k$; trained message-ablation policy $\hat{\pi}_i, \forall i \in [N]$,
2: **repeat**
3:     **for** $i = 1$ **to** $N$ **do**
4:         Receive $\mathbf{m}_{:\to i}$, $o_i$ and update history $\tau_i$
5:         Take $\widetilde{a}_i \leftarrow \widetilde{\pi}_i(\tau_i, \mathbf{m}_{:\to i})$ as in Def. 4.2.
6:     **end for**
7: **until** end of test

---

with ablation and aggregation during test time to defend against adversarial communication. The main idea is to collect all possible k-samples from $\mathcal{H}_k(\mathbf{m})$, and select an action voted by the majority of those k-samples. Specifically, we construct a *message-ensemble policy* $\widetilde{\pi} : \Gamma \times \mathcal{M}^{N-1} \to \mathcal{A}$ that outputs an action by aggregating the base actions produced by $\hat{\pi}$ on multiple k-samples (Line 5 in Algorithm 2). The construction of the message-ensemble policy depends on whether the agent's action space $\mathcal{A}$ is discrete or continuous, which is given below by Definition 4.2.

**Definition 4.2** (Message-Ensemble Policy). *For a message-ablation policy $\hat{\pi}$ with observation history $\tau$ and received message set $\mathbf{m}$, define the message-ensemble policy $\widetilde{\pi}$ as*

$$\widetilde{\pi}(\tau, \mathbf{m}) := \text{argmax}_{a \in \mathcal{A}} \sum\nolimits_{[\mathbf{m}]_k \in \mathcal{H}_k(\mathbf{m})} \mathbb{1}[\hat{\pi}(\tau, [\mathbf{m}]_k) = a], \qquad \textit{if } \mathcal{A} \textit{ is discrete, and} \quad (1)$$

$$\widetilde{\pi}(\tau, \mathbf{m}) := \text{Median}\{\hat{\pi}(\tau, [\mathbf{m}]_k) : [\mathbf{m}]_k \in \mathcal{H}_k(\mathbf{m})\}, \qquad \textit{if } \mathcal{A} \textit{ is continuous.} \quad (2)$$

Therefore, the message-ensemble policy $\widetilde{\pi}$ takes the action suggested by the consensus of all k-samples (majority vote in a discrete action space, and coordinate-wise median for a continuous action space). Different from model-ensemble methods (Kurutach et al., 2018; Harutyunyan et al., 2014) that learn multiple network models, we use the ensemble of messages, and only train a single policy network $\hat{\pi}$. Therefore, the *training process does not require extra computations*. We will show in the next section that, with some mild conditions, $\widetilde{\pi}$ under adversarial communications works similarly as the message-ablation policy $\hat{\pi}$ under all-benign communications.

## 4.2 ROBUSTNESS CERTIFICATES OF AME

In this section, we theoretically analyze the robustness of AME. During test time, at any step, let $\tau$ be the interaction history, $\mathbf{m}_{\text{benign}}$ be the unperturbed benign message set, and $\mathbf{m}_{\text{adv}}$ be the perturbed message set. Note that $\mathbf{m}_{\text{benign}}$ and $\mathbf{m}_{\text{adv}}$ both have $N - 1$ messages while they differ by up to $C$ messages. With the above notation, we define a set of actions rendered by purely benign k-samples in Definition 4.3. As the agent using a well-trained message-ablation policy is likely to take these actions in a non-adversarial environment, they can be regarded as "good" actions to take.

**Definition 4.3** (Benign Action Set $\mathcal{A}_{\text{benign}}$). *For the execution of the message-ablation policy $\hat{\pi}$ at any step, define $\mathcal{A}_{\text{benign}} \subseteq \mathcal{A}$ as a set of actions that $\hat{\pi}$ may select under benign k-samples.*

$$\mathcal{A}_{\text{benign}} := \cup_{[\mathbf{m}_{\text{benign}}]_k \in \mathcal{H}_k(\mathbf{m}_{\text{benign}})} \left\{ \hat{\pi}(\tau, [\mathbf{m}_{\text{benign}}]_k) \right\}. \quad (3)$$

Our robustness certificates are based on the intuition that the consensus of ablated messages is benign, under the condition that a consensus exists among benign messages. Next, we present the action certificates and reward certificates for discrete and continuous actions, respectively.

### 4.2.1 ROBUSTNESS CERTIFICATES FOR DISCRETE ACTION SPACE

We first characterize the condition for the existence of consensus. Intuitively, the following condition ensures that the action resulted by majority vote stands for the consensus of benign messages.

**Condition 4.4** (Dominating Benign Votes). *Denote the highest number of votes among all actions given message set $\mathbf{m}$ as $u_{\max}(\mathbf{m}) := \max_{a \in \mathcal{A}} \sum_{[\mathbf{m}]_k \in \mathcal{H}_k(\mathbf{m})} \mathbb{1}[\hat{\pi}(\tau, [\mathbf{m}]_k) = a]$ satisfies*

$$u_{\max}(\mathbf{m}) > \binom{N-1}{k} - \binom{N-1-C}{k} =: u_{\text{adv}}, \quad (4)$$

*where $u_{\text{adv}}$ is the number of votes that adversarial messages may affect (the number of k-samples that contain at least one adversarial message).*

**Remarks**. (1) This condition ensures the consensus has more votes than the votes that adversarial messages are involved in. Therefore, when $\widetilde{\pi}$ takes an action, there must exist some purely-benign k-samples voting for this action. (2) This condition is easy to satisfy when $C \ll N$ as $\binom{N-1}{k} \approx \binom{N-1-C}{k}$. (3) This condition can be easily checked during execution given an upper bound of $C$.

The following theorem suggests that under the above condition. the ensemble policy $\widetilde{\pi}$ always takes benign actions suggested by benign message combinations, no matter whether attacks exist.

**Theorem 4.5** (Action Certificate for Discrete Action Space). *For a perturbed message set $\mathbf{m}_{\text{adv}}$, if Condition 4.4 holds, the ensemble policy $\widetilde{\pi}$ in Equation (1) produces benign actions under $\mathbf{m}_{\text{adv}}$:*

$$\widetilde{a} = \widetilde{\pi}(\tau, \mathbf{m}_{\text{adv}}) \in \mathcal{A}_{\text{benign}}. \quad (5)$$

Then, we can further derive a reward certificate as introduced below.

**Reward Certificate for Discrete Action Space** When Condition 4.4 holds, the message-ensemble policy $\widetilde{\pi}$'s action in every step falls into a benign action set, and thus *its cumulative reward under adversarial communications is also guaranteed to be no lower than the worst natural reward the base policy $\hat{\pi}$ can get under random benign message subsets.* Therefore, adversarial communication under Assumption 3.1 cannot drop the reward of any agent trained with AME. The formal reward certificate is shown in Corollary C.1 in Appendix C.1.

### 4.2.2 ROBUSTNESS CERTIFICATES FOR CONTINUOUS ACTION SPACE

For a continuous action space, the following condition is needed, which can always be satisfied by the proper selection of ablation size $k$.

**Condition 4.6** (Dominating Benign Samples). *The ablation size $k$ of AME satisfies*

$$\binom{N-1-C}{k} > \frac{1}{2}\binom{N-1}{k}. \tag{6}$$

**Remarks.** (1) For the message set $\mathbf{m}_{\mathrm{adv}}$ that has up to $C$ adversarial messages, this condition implies that among all k-samples from $\mathbf{m}_{\mathrm{adv}}$, there are more purely benign k-samples than k-samples that contain adversarial messages. (2) Under Assumption 3.1, this condition *always has solutions* for $k$. (3) With an upper bound of $C$, one can choose $k$ as the largest solution to Equation (6).

**Theorem 4.7** (Action Certificate for Continuous Action Space). *Under Condition 4.6, the action $\widetilde{a} = \widetilde{\pi}(\tau, \mathbf{m}_{\mathrm{adv}})$ generated by the ensemble policy $\widetilde{\pi}$ defined in Equation* (2) *satisfies*

$$\widetilde{a} \in \mathsf{Range}(\mathcal{A}_{\mathrm{benign}}) := \{a : \forall 1 \leq l \leq L, \exists \underline{a}, \overline{a} \in \mathcal{A}_{\mathrm{benign}} \; s.t \; \underline{a}_l \leq a_l \leq \overline{a}_l\}. \tag{7}$$

Theoretically, $\mathsf{Range}(\mathcal{A}_{\mathrm{benign}})$ is a set of actions that are coordinate-wise bounded by base actions resulted from purely benign k-samples. In many practical problems, it is reasonable to assume that actions in $\mathsf{Range}(\mathcal{A}_{\mathrm{benign}})$ are relatively safe, especially when benign actions in $\mathcal{A}_{\mathrm{benign}}$ are concentrated. For instance, if benign message combinations have suggested driving at 40 mph or driving at 50 mph, then driving at 45 mph is also safe. More interpretations are in Appendix C.2. The above guarantee also leads to a bounded gap between natural reward and attacked reward.

**Reward Certificate for Continuous Action Space.** We can show that the difference between the cumulative reward of the message-ensemble policy $\widetilde{\pi}$ under adversarial communications and the natural reward of $\hat{\pi}$ is no larger than $\frac{\epsilon_R + \gamma V_{\max}\epsilon_P}{1-\gamma}$, where $V_{\max}$ is the maximal value of the agent, and $\epsilon_R, \epsilon_P$ are constants measuring the smoothness of environment dynamics wrt $\mathsf{Range}(\mathcal{A}_{\mathrm{benign}})$, formally defined in Appendix C.2. When the environment dynamics are smooth (agents do not immediately fail due to a single mistake) and the benign messages achieve consensus ($\mathsf{Range}(\mathcal{A}_{\mathrm{benign}})$ is concentrated), then $\epsilon_R$ and $\epsilon_P$ are small, and $\widetilde{\pi}$ does not suffer from drastic reward drop.

### 4.2.3 INTERPRETATION OF CONDITIONS AND HYPERPARAMETER

**How to Select Ablation Size $k$: Trade-off between Performance and Robustness.** The ablation size $k$ is an important hyperparameter for the guarantees to hold. In general, a smaller $k$ can tolerate a larger $C$, and a smaller $C$ can be defended by a wider range of $k$, as theoretically analyzed in Appendix F. However, a smaller $k$ restricts the power of information sharing, as the message-ablation policy can access fewer messages in one step. Therefore, the value of $k$ is related to the intrinsic trade-off between robustness and natural performance (Tsipras et al., 2019; Zhang et al., 2019), which suggests that seeking high natural performance may hurt robustness, and *achieving strong robustness may sacrifice some natural performance*. In practice, one can also train several message-ablation policies with different $k$'s during training, and later adaptively select a message-ablation policy to construct a message-ensemble policy during execution (decrease $k$ if higher robustness is needed).

**Worst-Case Adversaries.** Condition 4.4 and Condition 4.6 consider the worst-case scenario when the adversarial messages collaborate to dominate the consensus, which is intrinsically harsh for the victim — if the victim fails to identify or filter out the adversarial messages, its performance can be arbitrarily bad in the worst case. In contrast, although AME sacrifices some natural performance, its worst-case performance can be guaranteed. When these conditions do not hold, our algorithm can still achieve strong robustness under sub-optimal attacks, as verified in experiments.

**Information Redundancy Leads to Higher Robustness.** Intuitively, AME prefers benign messages to contain certain amount of redundant information (consensus), which is similar to a multi-factor authentication in security-critical applications. This allows AME to tolerate arbitrary perturbations without assumptions such as a bounded $\ell_p$-norm. More importantly, AME provides a way to establish and utilize such consensus, by explicitly setting $k$ and checking the required conditions.

### 4.3 SCALING UP: ENSEMBLE WITH PARTIAL SAMPLES

So far we have discussed the proposed AME defense and the constructed ensemble policy that aggregates all $\binom{N-1}{k}$ number of k-samples out of $N-1$ messages. However, if $N$ is large, sampling all $\binom{N-1}{k}$ combinations of message subsets could be expensive. In this case, a smaller number of

k-samples can be used. That is, given a sample size $0 < D \leq \binom{N-1}{k}$, we randomly select $D$ number of k-samples from $\mathcal{H}_k(\mathbf{m})$ (without replacement), and then we aggregate the message-ablation policy's decisions on selected k-samples. In this way, we define a partial-sample version of the ensemble policy, namely *D-ensemble policy* $\widetilde{\pi}_D$. In this case, the robustness guarantee holds with high probability that increases as $D$ gets larger. For example, in a continuous action space, we show that Equation (7) holds with probability at least

$$p_D = \frac{\sum_{j=\lfloor \frac{D}{2} \rfloor + 1}^{D} \binom{n_2}{j} \binom{n_1 - n_2}{D - j}}{\binom{n_1}{D}}. \tag{8}$$

Formal definitions and full theoretical analysis of this partial-sample variant of AME are in Appendix C.3. Our experiments show that AME works well with a relatively small $D$.

## 5 RELATED WORK

**Certifiable Defenses.** For more reliable application of deep learning, many approaches have been developed to certify the performance of neural networks, including semidefinite programming-based defenses (Raghunathan et al., 2018a;b), convex relaxation of neural networks (Gowal et al., 2019; Zhang et al., 2018; Wong & Kolter, 2018; Zhang et al., 2020a; Gowal et al., 2018), randomized smoothing of a classifier (Cohen et al., 2019; Hayes, 2020), etc. Most existing works focus on the $\ell_p$ threat model where the perturbation is small in $\ell_p$ norm, while we consider a different and practical threat model as discussed in Section 2.

**Adversarial Robustness of RL Agents.** Appendix A introduces existing adversarial attacks on single-agent and multi-agent problems. To improve the robustness of agents, adversarial training (i.e., introducing adversarial agents to the system during training (Pinto et al., 2017; Phan et al., 2021; Zhang et al., 2021; Sun et al., 2022)) and network regularization (Zhang et al., 2020b; Shen et al., 2020; Oikarinen et al., 2021) are empirically shown to be effective under $\ell_p$ attacks, although such robustness is not theoretically guaranteed. In an effort to certify RL agents' robustness, some approaches (Lütjens et al., 2020; Zhang et al., 2020b; Oikarinen et al., 2021; Fischer et al., 2019) apply network certification tools to bound the Q networks. Kumar et al. (2021) and Wu et al. (2022) apply randomized smoothing (Cohen et al., 2019) to RL for provable robustness.

**Adversarial Attacks and Defenses in CMARL.** Appendix A discusses literature of learning effective communication among agents (Foerster et al., 2016; Sukhbaatar et al., 2016; Das et al., 2019). Note that the focus of this paper is defending against adversarial perturbations on existing communications, which is orthogonal to the concrete communication strategy. Recently, the existence of adversarial communication in MARL has attracted increasing attention. Blumenkamp & Prorok (2020) show that in a cooperative game, communication policies of some self-interest agents can hurt other agents' performance. To achieve robust communication, Mitchell et al. (2020) adopt a Gaussian Process-based probabilistic model to compute the posterior probabilities that whether each partner is truthful. Tu et al. (2021) investigate the vulnerability of multi-agent autonomous systems against $\ell_p$ communication attacks on vision tasks. Xue et al. (2022) propose to learn an anomaly detector and a message reconstructor to recover the true messages, and train two populations of defenders and attackers to improve the generalizability of defense. But in our formulation, the attacker may arbitrarily replace the messages so that recovering the true message is infeasible.

To the best of our knowledge, our AME is the first certifiable defense in MARL against communication attacks. Moreover, we consider a strong threat model where up to half of the communication messages can be arbitrarily corrupted, capturing many realistic types of attacks.

## 6 EMPIRICAL STUDY

In this section, we verify the robustness of our AME in multiple different CMARL environments against various communication attack algorithms. Then, we conduct hyperparameter tests for the ablation size $k$ and the sample size $D$ (for the variant of AME introduced in Section 4.3).

**Environments.** To evaluate the effectiveness of AME, we consider the following four environments that cover various environment and communication settings.
• *FoodCollector (discrete/continuous action, pre-defined communication)*: $N = 9$ agents with different colors search for foods with the same colors. Agents use their limited-range sensors to observe the surrounding objects, while communicating with each other to share their recently observed

food locations. The action space can be either discrete (9 moving directions and 1 no-move action), or continuous (2-dimensional vector denoting acceleration). Agents are penalized for having leftover foods at every step, so they seek to find all foods as fast as possible.

● *InventoryManager (continuous action, pre-defined communication)*: $N = 10$ cooperative distributor agents carry inventory for 3 products. There are 300 buyers sharing an underlying demand distribution. At every step, each buyer requests products from a randomly selected distributor. Then, each distributor agent observes random demand requests, takes restocking actions to adjust its own inventory, and communicates with others by sending its demand observations. Distributors are penalized for mismatches between their inventory and the demands.

● *MARL-MNIST (discrete action, learned communication)*(Mousavi et al., 2019): $N = 9$ agents aim to classify images in MNIST within a limited time horizon. At every step, each agent observes a patch of the image, and can take an action to move to an adjacent patch. Agents are allowed to exchange information (encoded local beliefs) with other agents to update their own beliefs.

● *Traffic Junction (discrete action, learned communication) (Sukhbaatar et al., 2016)*: $N = 10$ cars move along potentially intersecting routes on one or more road junctions. Each car has a limited visibility but is free to communicate with all other cars. The goal is pass the road without collision. The detailed description and policy implementation for each task are in Appendix E.1 and E.2.

**Baselines.** Based on the same policy learning paradigm for each environment, we compare our AME with two defense baselines: **(1)** *(Vanilla)*: vanilla training without defense, which learns a policy based on all benign messages. **(2)** *(AT)* adversarial training as in Zhang et al. (2021), which alternately trains an adaptive RL attacker and an agent. During training and defending, we set the ablation size AME as $k = 2$, the largest solution to Equation (6) for $C = 2$. For AT, we train the agent against $C = 2$ learned adversaries. More implementation details are provided in Appendix E.2.

**Evaluation Metrics.** To evaluate the effectiveness of our defense strategy, we test the performance of the trained policies under no attack and under various values of $C$ (for simplicity, we refer to $C$ as the *number of adversaries*). Different from previous work (Blumenkamp & Prorok, 2020) where the adversarial agent disrupts all the other agents, we consider the case where the attacker deliberately misleads a selected victim, which could evaluate the robustness of the victim under the harshest attacks. We report the victim's local reward under the following two types of attack methods:

**(1)** *Heuristic attack* that perturbs messages based on heuristics. In FoodCollector, MARL-MNIST and Traffic Junction, we consider randomly generated messages within the valid range of communication messages. Note that this is already a strong attack since a random message could be arbitrarily far from the original message. For InventoryManager where messages have clear physical meanings (demand of buyers), we consider 3 realistic attacks: Perm-Attack, Swap-Attack, Flip-Attack, which permute, swap or flip the demand observations, respectively, as detailed in Appendix E.2.2.

**(2)** *Learned adaptive attack* that learns the strongest/worst adversarial communication with an RL algorithm to minimize the victim's reward (a white-box attacker which knows the victim's reward). The learned attack can strategically mislead the victim. As shown in prior works (Zhang et al., 2020b;

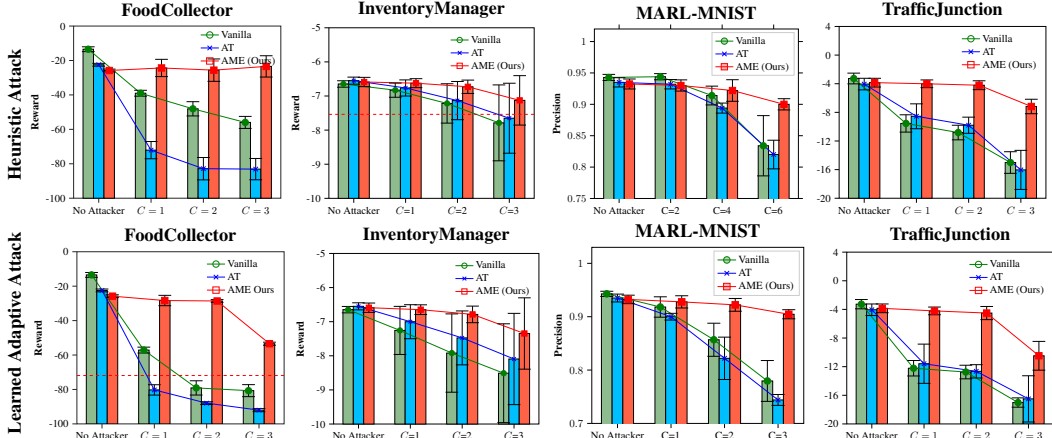

**Figure 2:** Rewards of our AME and baselines in all environments, under no attacker and varying numbers of adversaries for adaptive and various non-adaptive attacks. The dashed red lines stand for the average performance of a non-communicative agent. Results are averaged over 5 random seeds. Our AME outperforms all baseline methods in all tasks, and stays robust for varying number of adversaries (denoted by $C$).

2021; Sun et al., 2022), the theoretically optimal attack (which minimizes the victim's reward) can be formed as an RL problem and learned by RL algorithms. Therefore, we can regard this attack as a worst-case attack for the victim agents. More details are in Appendix E.2.

**Experiment Results.** The major results are shown in Figure 2, where we present the discrete-action FoodCollector, and use Swap-Attack as the heuristic attack for InventoryManager. More results including continuous-action FoodCollector and other heuristic attacks for InventoryManager are put in Appendix E.3. We can see that the rewards of Vanilla and AT drastically drop under attacks. Under strong adaptive attackers, Vanilla and AT sometimes perform worse than a non-communicative agent shown by dashed red lines, which suggests that communication can be a double-edged sword. Although AT is usually effective for $\ell_p$ attacks (Zhang et al., 2021), we find that AT does not achieve better robustness than Vanilla, since it can not adapt to arbitrary perturbations to several messages. (More analysis of AT is in Appendix E.3.5.) In contrast, *AME can utilize benign communication well while being robust to adversarial communication.*

We use $k = 2$ for our AME, which in theory provides performance guarantees against up to $C = 2$ adversaries for $N = 9$ and $N = 10$. We can see that the reward of AME under $C = 1$ or $C = 2$ is similar to its reward under no attack, matching our theoretical analysis. Even under 3 adversaries where the theoretical guarantees no longer hold, AME still obtains superior performance compared to Vanilla and AT. Therefore, *AME makes agents robust under varying numbers of adversaries.*

**Discussion on Ablation Size $k$** We demonstrate AME's natural reward and attacked reward in discrete-action FoodCollector under $C = 2$ adversaries with *all possible values of $k$ ranging from 1 to $N - 1$* in Figure 3(left). Results in other environments are similar and are put in Appendix E.3.4. The green curve shows that the natural performance of AME increases as $k$ gets larger, where $k = 1$ is the most conservative version of AME, and $k = N - 1$ degenerates to the vanilla policy without message ablation. It

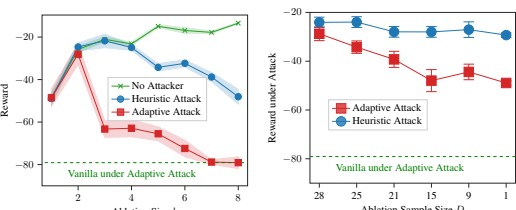

**Figure 3:** Natural and robust performance of AME with various values of **(left)** ablation size $k$ and **(right)** sample size $D$, in discrete-action FoodCollector under $C = 2$.

is intuitive because a larger $k$ allows the agent to gather more information from others. Although a larger $k$ is beneficial in a clean environment, gathering information without defense makes the agent more vulnerable to communication attacks. The red and blue curves show that the reward of agents decreases under attacks when $k$ gets larger, especially when attacks are adaptive. Therefore, *increasing ablation size $k$ trades off robustness for natural performance*, matching the analysis in Section 4.2.3. As the largest solution to Equation (6) in Condition 4.6, ablation size $k = 2$ achieves a good balance between performance and robustness. Even when $k > 2$ which breaks Condition 4.6, AME is still more robust than baselines, showing the flexibility of AME.

**Discussion on for Sample Size $D$** We evaluate the partial-sample variant of AME introduced in Section 4.3 with $D$ varying from $\binom{N-1}{k}$ (ensemble of all message combinations) to 1 (randomly take one k-sample), under $k = 2$ and $C = 2$. Figure 3(right) demonstrates the performance of different $D$'s in discrete-action FoodCollector, and Appendix E.3 shows results in other environments. As $D$ goes down, AME obtains lower reward under attackers, but it is still significantly more robust than baseline methods. Note that $D = 1$ is equivalent to executing the message-ablation policy without ensemble, which is robust to heuristic attacks but less robust to adaptive attacks than the original AME, verifying the effectiveness of message ensemble.

## 7 CONCLUSION AND DISCUSSION

This paper proposes a defense framework AME, which is certifiably robust against multiple arbitrarily perturbed adversarial communication messages, for any existing communication protocol. Our proposed ablation and ensemble method can be extended to robustify other decision makers which takes in multiple possibly-untrustworthy information sources. The limitation of AME is the requirement of several conditions. Although these conditions can be quantified and checked in practice, our future work aims to relax these conditions, or to learn a communication policy satisfying these conditions. AME utilizes the information redundancy of communication, which can be achieved by the learning of a communication policy in many tasks, as an extension of the current method.

## 8 REPRODUCIBILITY STATEMENT

Our implementation of the AME algorithm and the FoodCollector environment are available at https://github.com/umd-huang-lab/cmarl_ame.git.

## ACKNOWLEDGEMENTS

This work is supported by JPMorgan Chase & Co., National Science Foundation NSF-IIS-FAI Award 2147276, Office of Naval Research, Defense Advanced Research Projects Agency Guaranteeing AI Robustness against Deception (GARD), and Adobe, Capital One and JP Morgan faculty fellowships.

**Disclaimer** This paper was prepared for informational purposes in part by the Artificial Intelligence Research group of JPMorgan Chase & Coànd its affiliates ("JP Morgan"), and is not a product of the Research Department of JP Morgan. JP Morgan makes no representation and warranty whatsoever and disclaims all liability, for the completeness, accuracy or reliability of the information contained herein. This document is not intended as investment research or investment advice, or a recommendation, offer or solicitation for the purchase or sale of any security, financial instrument, financial product or service, or to be used in any way for evaluating the merits of participating in any transaction, and shall not constitute a solicitation under any jurisdiction or to any person, if such solicitation under such jurisdiction or to such person would be unlawful.

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

# Appendix: Certifiably Robust Policy Learning against Adversarial Multi-Agent Communication

## A  ADDITIONAL RELATED WORK

**Communication in MARL**  Communication is crucial in solving collaborative MARL problems. There are many existing studies learning communication protocols across multiple agents. Foerster et al. (2016) are the first to learn differentiable communication protocols that is end-to-end trainable across agents. Another work by Sukhbaatar et al. (2016) proposes an efficient permutation-invariant centralized learning algorithm which learns a large feed-forward neural network to map inputs of all agents to their actions. Attention mechanisms are also proven to be effective in learning communication (Jiang & Lu, 2018; Rangwala & Williams, 2020). Some recent works (Yuan et al., 2022; Kim et al., 2021) propose new communication schemes that models other agents' actions or intentions. It is also important to communicate selectively, since some communication may be less informative or unnecessarily expensive. To tackle this challenge, Das et al. (2019) propose an attention mechanism for agents to adaptively select which other agents to send messages to. Liu et al. (2020) introduce a handshaking procedure so that the agents communicate only when needed.

This paper proposes a certifiably robust algorithm against perturbations on communications, which is orthogonal to the concrete communication strategy. Our AME can be combined with the above communication MARL methods to improve their robustness.

**Other Attacks in RL and MARL**  Adversarial attacks and defenses in RL systems have recently attracted more and more attention, and are considered in many different scenarios. The majority of related work on adversarial RL focuses on *directly attacking a victim* by perturbing its observations (Huang et al., 2017; Oikarinen et al., 2021; Zhang et al., 2020b; Sun et al., 2022; Korkmaz, 2021; 2022; 2023; Liang et al., 2022) or actions (Tessler et al., 2019; Pinto et al., 2017). However, an attacker may not have direct access to the specific victim's observation or action. In this case, *indirect attacks via other agents* can be an alternative. Gleave et al. (2020); Liu et al. (2022) propose to attack the victim by changing the other agent's actions. Therefore, even if the victim agent has well-protected sensors, the attacker can still influence it by manipulating other under-protected agents. But the intermediary agent whose actions are altered will obtain sub-optimal reward, which makes the attack noticeable and less stealthy. In contrast, we consider the scenario where an attacker alters the communication messages sent from some other agents to the victim without changing the behaviors of these agents. In this case, it is relatively hard for the victim to identify the attacks.

Training-time attacks, or poisoning (Behzadan & Munir, 2017; Huang & Zhu, 2019; Rakhsha et al., 2020; Sun et al., 2021) propose to manipulate the training data such that the agent learns a bad or target policy, different from evasion attacks that deprave a well-trained policy. Investigation on how training-time communication perturbations influence the learned policies would be an interesting future direction.

## B  ALGORITHM PSEUDOCODE

Algorithm 3 and Algorithm 4 demonstrate the procedures of AME in training and defending phases, respectively.

## C  ADDITIONAL THEORETICAL DEFINITIONS AND ANALYSIS

**Relation between Communication Attack and $\ell_0$ Observation Attack**  The communication threat model described in Section 3 is analogous to a constrained $\ell_0$ attack on policy inputs. When agent $i$ is attacked, the input space of its acting policy $\pi_i$ is $\mathcal{X}_i := \Gamma_i \times \mathcal{M}^{N-1}$. Therefore, when $C$ communication messages are corrupted, the original input $x_i$ gets perturbed to $\tilde{x}_i$. Let $d$ be the dimension of a communication message, then $x_i$ and $\tilde{x}_i$ differ by up to $dC$ dimensions, which is similar to an $\ell_0$ attack constrained in certain dimensions.

---

**Algorithm 3** Training Phase of AME

---

1: **Input:** ablation size $k$
2: Initialize $\hat{\pi}_i$ for every agent $i \in [N]$.
3: **repeat**
4:     **for** $i = 1$ **to** $N$ **do**
5:         Receive a list of messages $\mathbf{m}_{:\to i}$, get local observation $o_i$ and update interaction history $\tau_i$
6:         Randomly sample $[\mathbf{m}_{:\to i}]_k \sim \text{Uniform}(\mathcal{H}_k(\mathbf{m}_{:\to i}))$
7:         Take action based on $\tau_i$ and the k-sample $[\mathbf{m}_{:\to i}]_k$, i.e., $a_i \leftarrow \hat{\pi}_i(\tau_i, [\mathbf{m}_{:\to i}]_k)$
8:         Update the replay buffer and policy $\hat{\pi}_i$
9:     **end for**
10: **until** end of training
11: **Output:** message-ablation policy $\hat{\pi}_i, \forall i \in [N]$

---

---

**Algorithm 4** Defending Phase of AME

---

1: **Input:** ablation size $k$; trained message-ablation policy $\hat{\pi}_i, \forall i \in [N]$,
2: **repeat**
3:     **for** $i = 1$ **to** $N$ **do**
4:         Receive a list of messages $\mathbf{m}_{:\to i}$ with at most $C$ malicious messages, get local observation $o_i$ and update interaction history $\tau_i$
5:         Take $\tilde{a}_i \leftarrow \tilde{\pi}_i(\tau_i, \mathbf{m}_{:\to i})$, where $\tilde{\pi}_i$ is the message-ensemble policy defined with $\hat{\pi}$ by Equation (1) for discrete $\mathcal{A}_i$, and Equation (2) for continuous $\mathcal{A}_i$
6:     **end for**
7: **until** end of test

---

### C.1   ADDITIONAL ANALYSIS OF AME IN DISCRETE ACTION SPACE

**Reward Certificate**   Theorem 4.5 justifies that under sufficient majority votes, the message-ensemble policy $\tilde{\pi}$ ignores the malicious messages in $\mathbf{m}_{\text{adv}}$ and executes a benign action that is suggested by some benign message combinations, even when the malicious messages are not identified. It is important to note that, when the message-ensemble policy $\tilde{\pi}$ selects an action $\tilde{a}$, there must exist at least one purely benign k-sample that let the message-ablation policy $\hat{\pi}$ produce $\tilde{a}$. Therefore, as long as $\hat{\pi}$ can obtain high reward with randomly selected benign k-samples, $\tilde{\pi}$ can also obtain high reward with ablated adversarial communication due to its design.

Specifically, we consider a specific agent with message-ablation policy $\hat{\pi}$ and message-ensemble policy $\tilde{\pi}$ (suppose other agents are executing fixed policies). Let $\nu : \mathcal{M}^{N-1} \to \mathcal{M}^{N-1}$ be an attack algorithm that perturbs at most $C$ messages in a message set. Let $\zeta \sim Z(P, \hat{\pi})$ be a trajectory of policy $\hat{\pi}$ under no attack, i.e., $\zeta = (o^{(0)}, \mathbf{m}^{(0)}, a^{(0)}, r^{(0)}, o^{(1)}, \mathbf{m}^{(1)}, a^{(0)}, r^{(0)}, \cdots)$. (Recall that a message-ablation policy $\hat{\pi}$ takes in a random size-$k$ subset of $\mathbf{m}^{(t)}$ and outputs action $a^{(t)}$.) When there exists attack with $\nu$, let $\zeta_\nu \sim Z(P, \tilde{\pi}; \nu)$ be a trajectory of policy $\tilde{\pi}$ under communication attacks, i.e., $\zeta_\nu = (o^{(0)}, \nu(\mathbf{m}^{(0)}), a^{(0)}, r^{(0)}, o^{(1)}, \nu(\mathbf{m}^{(1)}), a^{(0)}, r^{(0)}, \cdots)$. For any trajectory $\zeta$, let $r(\zeta)$ be the discounted cumulative reward of this trajectory.

With the above notations, we propose the following reward certificate.

**Corollary C.1** (Reward Certificate for Discrete Action Space). *When Condition 4.4 holds at every step of execution, the cumulative reward of ensemble policy $\tilde{\pi}$ defined in Equation (1) under adversarial communication is no lower than the lowest cumulative reward that the ablation policy $\hat{\pi}$ can obtain with randomly selected k-samples under no attacks, i.e.,*

$$\min_{\zeta_\nu \sim Z(P, \tilde{\pi}; \nu)} r(\zeta_\nu) \geq \min_{\zeta \sim Z(P, \hat{\pi})} r(\zeta), \tag{9}$$

*for any attacker $\nu$ satisfying Assumption 3.1 ($C < \frac{N-1}{2}$).*

**Remarks.** (1) The certificate holds for any attack algorithm $\nu$ with $C < \frac{N-1}{2}$. (2) The message-ablation policy $\hat{\pi}$ has extra randomness from the sampling of k-samples. That is, at every step, $\hat{\pi}$ takes a uniformly randomly selected k-sample from $\mathcal{H}_k(\mathbf{m})$. Therefore, the $\min_\zeta$ in the RHS considers the worst-case message sampling in the clean environment without attacks. Since $\widetilde{\pi}$ always takes actions selected by some purely-benign message combinations, the trajectory generated by $\widetilde{\pi}$ can also be produced by the message-ablation policy. (3) Note that the RHS ($\min_{\zeta \sim Z(P,\hat{\pi})} r(\zeta)$) can be approximately estimated by executing $\hat{\pi}$ during training time, so that the test-time performance of $\widetilde{\pi}$ is guaranteed to be no lower than this value, even if there are up to $C$ corrupted messages at every step.

Appendix D provides detailed proofs for the above theoretical results.

## C.2 Additional Analysis of AME in Continuous Action Space

Note that Theorem 4.7 certifies that the selected action is in a set of actions that are close to benign actions $\mathsf{Range}(\mathcal{A}_{\text{benign}})$, but does not make any assumption on this set. Next we interpret this result in details.

**How to Understand** $\mathsf{Range}(\mathcal{A}_{\text{benign}})$**?** Theoretically, $\mathsf{Range}(\mathcal{A}_{\text{benign}})$ is a set of actions that are coordinate-wise bounded by base actions resulted from purely benign k-samples. In many practical problems, it is reasonable to assume that actions in $\mathsf{Range}(\mathcal{A}_{\text{benign}})$ are relatively safe, especially when benign actions in $\mathcal{A}_{\text{benign}}$ are concentrated. The following examples illustrate some scenarios where actions in $\mathsf{Range}(\mathcal{A}_{\text{benign}})$ are relatively good.

1. If the action denotes the price a seller sells its product in a market, and the communication messages are the transaction signals in an information pool, then $\mathsf{Range}(\mathcal{A}_{\text{benign}})$ is a price range that is determined based on purely benign messages. Therefore, the seller will set a reasonable price without being influenced by a malicious message.

2. If the action denotes the driving speed, and benign message combinations have suggested driving at 40 mph or driving at 50 mph, then driving at 45 mph is also safe.

3. If the action is a vector denoting movements of all joints of a robot (as in many MuJoCo tasks), and two slightly different joint movements are suggested by two benign message combinations, then an action that does not exceed the range of the two benign movements at every joint is likely to be safe as well.

The above examples show by intuition why the message-ensemble policy can be regarded as a relatively robust policy. However, in extreme cases where there exists "caveat" in $\mathsf{Range}(\mathcal{A}_{\text{benign}})$, taking an action in this set may also be unsafe. To quantify the influence of $\mathsf{Range}(\mathcal{A}_{\text{benign}})$ on the long-term reward, we next analyze the cumulative reward of the message-ensemble policy in the continuous-action case.

**How Does** $\mathsf{Range}(\mathcal{A}_{\text{benign}})$ **Lead to A Reward Certificate?** Different from the discrete-action case, the message-ensemble policy $\widetilde{\pi}$ in a continuous action space may take actions not in $\mathcal{A}_{\text{benign}}$ such that it generates trajectories not seen by the message-ablation policy $\hat{\pi}$. However, since the action of $\widetilde{\pi}$ is guaranteed to stay in $\mathsf{Range}(\mathcal{A}_{\text{benign}})$, we can bound the difference between the value of $\widetilde{\pi}$ and the value of $\hat{\pi}$, and how large the different is depends on some properties of $\mathsf{Range}(\mathcal{A}_{\text{benign}})$.

Concretely, Let $R$ and $P$ be the reward function and transition probability function of the current agent when the other agents execute fixed policies. So $R(s,a)$ is the immediate reward of taking action $a$ at state $s$, and $P(s'|s,a)$ is the probability of transitioning to state $s'$ from $s$ by taking action $a$. (Note that $s$ is the underlying state which may not be observed by the agent.)

**Definition C.2** (Dynamics Discrepancy of $\hat{\pi}$). *A message-ablation policy $\hat{\pi}$ is called $\epsilon_R, \epsilon_P$-discrepant if $\epsilon_R$, $\epsilon_P$ are the smallest values such that for any $s \in \mathcal{S}$ and the corresponding benign action set $\mathcal{A}_{\text{benign}}$, we have $\forall a_1, a_2 \in \mathsf{Range}(\mathcal{A}_{\text{benign}})$,*

$$|R(s,a_1) - R(s,a_2)| \leq \epsilon_R, \tag{10}$$

$$\int |P(s'|s,a_1) - P(s'|s,a_2)|\mathrm{d}s' \leq \epsilon_P. \tag{11}$$

**Remarks.** (1) Equation (11) is equivalent to that the total variance distance between $P(\cdot|s, a_1)$ and $P(\cdot|s, a_2)$ is less than or equal to $\epsilon_P/2$. (2) For any environment with bounded reward, $\epsilon_R$ and $\epsilon_P$ always exist.

Definition C.2 characterizes how different the local dynamics of actions in $\mathsf{Range}(\mathcal{A}_{\mathrm{benign}})$ are, over all possible states. If $\mathsf{Range}(\mathcal{A}_{\mathrm{benign}})$ is small and the environment is relatively smooth, then taking different actions within this range will not result in very different future rewards. The theorem below shows a reward certificate for the message-ensemble policy $\widetilde{\pi}$.

**Theorem C.3** (Reward Certificate for Continuous Action Space)**.** *Let $V^{\hat{\pi}}(s)$ be the clean value (discounted cumulative reward) of $\hat{\pi}$ starting from state $s$ under no attack; let $\tilde{V}_\nu^{\widetilde{\pi}}(s)$ be the value of $\widetilde{\pi}$ starting from state $s$ under attack algorithm $\nu$, where $\nu$ satisfies Assumption 3.1; let $k$ be an ablation size satisfying Condition 4.6. If $\hat{\pi}$ is $\epsilon_R,\epsilon_P$-discrepant, then for any state $s \in \mathcal{S}$, we have*

$$\min_\nu \tilde{V}_\nu^{\widetilde{\pi}}(s) \geq V^{\hat{\pi}}(s) - \frac{\epsilon_R + \gamma V_{\max}\epsilon_P}{1 - \gamma}, \tag{12}$$

*where $V_{\max} := \sup_{s,\pi} |V^\pi(s)|$.*

**Remarks.** (1) The certificate holds for any attack algorithm $\nu$ with $C < \frac{N-1}{2}$. (2) If $\epsilon_R$ and $\epsilon_P$ are small, then the performance of message-ensemble policy $\widetilde{\pi}$ under attacks is similar to the performance of the message-ablation policy $\hat{\pi}$ under no attack.

It is important to note that $\epsilon_R$ and $\epsilon_P$ are intrinsic properties of $\hat{\pi}$, independent of the attacker. Therefore, one can approximately measure $\epsilon_R$ and $\epsilon_P$ during training. Similar to Condition 4.4 required for a discrete action space, the gap between the attacked reward of $\widetilde{\pi}$ and the natural reward of $\hat{\pi}$ depends on how well the benign messages are reaching a consensus. (Smaller $\epsilon_R$ and $\epsilon_P$ imply that the actions in $\mathcal{A}_{\mathrm{benign}}$ are relatively concentrated and the environment dynamics are relatively smooth.)

Moreover, one can optimize $\hat{\pi}$ during training such that $\epsilon_R$ and $\epsilon_P$ are as small as possible, to further improve the robustness guarantee of $\widetilde{\pi}$. This can be a future extension of this work.

Technical proofs of all theoretical results can be found in Appendix D.

## C.3   EXTENSION OF AME WITH PARTIAL SAMPLES

As motivated in Section 4.3, our AME can be extended to a partial-sample version, where the ensemble policy is constructed by $D$ instead of $\binom{N-1}{k}$ samples. Let $\mathcal{H}_{k,D}(\mathbf{m})$ be a subset of $\mathcal{H}_k(\mathbf{m})$ that contains $D$ random k-samples from $\mathcal{H}_k(\mathbf{m})$. Then the $D$-ensemble policy $\pi_D$ is defined as

$$\widetilde{\pi}_D(\tau, \mathbf{m}) := \operatorname{argmax}_{a\in\mathcal{A}} \sum_{[\mathbf{m}]_k \in \mathcal{H}_{k,D}(\mathbf{m})} \mathbb{1}[\hat{\pi}(o, [\mathbf{m}]_k) = a], \tag{13}$$

for a discrete action space, and

$$\tilde{\pi}_D(\tau, \mathbf{m}) = \mathsf{Median}\{\hat{\pi}(\tau, [\mathbf{m}]_k)\}_{[\mathbf{m}]_k \in \mathcal{H}_{k,D}(\mathbf{m})}. \tag{14}$$

for a continuous action space.

In the partial-sample version of AME, we can still provide high-probability robustness guarantees.

For notation simplicity, let $n_1 = \binom{N-1}{k}$, $n_2 = \binom{N-C-1}{k}$. Define the majority vote as

$$u_{\max}(\mathbf{m}) := \max_{a\in\mathcal{A}} \sum_{[\mathbf{m}]_k \in \mathcal{H}_{k,D}(\mathbf{m})} \mathbb{1}[\hat{\pi}(\tau, [\mathbf{m}]_k) = a]. \tag{15}$$

The following theorem shows a general guarantee for $D$-ensemble policy in a discrete action space.

**Theorem C.4** (General Action Guarantee for Discrete Action Space)**.** *Given an arbitrary sample size $0 < D \leq \binom{N-1}{k}$, for the $D$-ensemble policy $\widetilde{\pi}_D$ defined in Equation (13), Equation (5) holds deterministically if the majority vote $u_{\max}(\mathbf{m}_{\mathrm{adv}}) > n_1 - n_2$. Otherwise it holds with probability at least*

$$p_D = \frac{\sum_{j=0}^{u_{\max}(\mathbf{m}_{\mathrm{adv}})-1} \binom{n_1-n_2}{j}\binom{n_2}{D-j}}{\binom{n_1}{D}}. \tag{16}$$

Note that Theorem 4.5 is a special case of Theorem C.4, since it assumes $u_{\max}(\mathbf{m}_{\mathrm{adv}}) > n_1 - n_2$.

Theorem C.5 below further shows the theoretical result for a continuous action space.

**Theorem C.5** (General Action Guarantee for Continuous Action Space). *Given an arbitrary sample size $0 < D \leq \binom{N-1}{k}$, for the D-ensemble policy $\widetilde{\pi}_D$ defined in Equation* (14) *with an ablation size $k$ satisfying Condition 4.6, Equation* (7) *holds with probability at least*

$$p_D = \frac{\sum_{j=\tilde{D}}^{D} \binom{n_2}{j}\binom{n_1-n_2}{D-j}}{\binom{n_1}{D}}, \tag{17}$$

*where $\tilde{D} = \lfloor \frac{D}{2} \rfloor + 1$.*

The larger $D$ is, the higher the probability $p_D$ is, the more likely that the message-ensemble policy selects an action in $\mathsf{Range}(\mathcal{A}_{\mathrm{benign}})$. In Theorem C.5, when $D = \binom{N-1}{k}$, the probability $p_D$ is 1 and the result matches Theorem 4.7.

Technical proofs of all theoretical results can be found in Appendix D.

# D  TECHNICAL PROOFS

For the simplicity of the proof, we make the following definition.

**Definition D.1.** *(Purely Benign k-sample and contaminated k-sample) A k-sample $[\mathbf{m}]_k \in \mathcal{H}_k(\mathbf{m})$ is purely benign if every message in $[\mathbf{m}]_k$ comes from a benign agent and is unperturbed. A k-sample $[\mathbf{m}]_k \in \mathcal{H}_k(\mathbf{m})$ is contaminated if there exists some message in $[\mathbf{m}]_k$ that is perturbed.*

For notation simplicity, let $n_1 := |\mathcal{H}_k(\mathbf{m})| = \binom{N-1}{k}$ be the total number of k-samples from a message set $\mathbf{m}$. Note that the total number of purely benign k-samples is $n_2 := \binom{N-C-1}{k}$, and the total number of contaminated k-samples is $n_1 - n_2 = \binom{N-1}{k} - \binom{N-C-1}{k}$.

## D.1  PROOFS IN DISCRETE ACTION SPACE

**Action Certificates**  We first prove the action certificates in the discrete action. For notation simplicity, we slightly abuse notation and use $u_{\max}$ to denote $u_{\max}(\mathbf{m}_{\mathrm{adv}})$. Note that Theorem 4.5 is a special case of Theorem C.4 ($u_{\max} > n_1 - n_2$ and $D = \binom{N-1}{k}$), so we first prove the general version Theorem C.4 and then Theorem 4.5 holds as a result.

*Proof of Theorem C.4 and Theorem 4.5.* The majority voted action $\tilde{a}$ is a benign action, i.e., $\tilde{a} \in \mathcal{A}_{\mathrm{benign}}$, if the ablation policy $\hat{\pi}$ renders action $\tilde{a}$ for at least one purely benign k-sample. If $u_{\max} > n_1 - n_2$, since $n_1 - n_2$ is exactly the total number of contaminated k-samples, then it is guaranteed that there is at least one purely benign k-sample for which $\hat{\pi}$ renders $\tilde{a}$. Thus, $\tilde{a} \in \mathcal{A}_{\mathrm{benign}}$, and Theorem 4.5 holds.

On the other hand, if $u_{\max} \leq n_1 - n_2$, then in order for $\tilde{a}$ to be in $\mathcal{A}_{\mathrm{benign}}$, among the $u_{\max}$ k-samples resulting in $\tilde{a}$ there can be at most $u_{\max} - 1$ contaminated k-samples. There are $\sum_{j=0}^{u_{\max}-1} \binom{n_1-n_2}{j}\binom{n_2}{D-j}$ such combinations in terms of the sampling of $D$, and the total number of combinations are $\binom{n_1}{D}$. Therefore, we get Equation (16).

$\square$

**Reward Certificate.**  Next, following Theorem 4.5, we proceed to prove the reward certificate.

*Proof of Corollary C.1.* Based on the definition of benign action set $\mathcal{A}_{\mathrm{benign}}$, $\widetilde{\pi}$ selects an action $\widetilde{a}$ if and only if there exists a purely benign k-sample $[\mathbf{m}]_k \in \mathcal{H}_k(\mathbf{m})$ such that the message-ablation policy $\hat{\pi}$ selects $\widetilde{a} = \hat{\pi}(\tau, [\mathbf{m}]_k)$. Therefore, for any trajectory generated by $\widetilde{\pi}$ under attacks, there is a trajectory of $\hat{\pi}$ with a list of k-samples $[\mathbf{m}]_k^{(1)}, [\mathbf{m}]_k^{(2)}, \cdots$ that renders the same cumulative reward under no attack.

$\square$

## D.2 PROOFS IN CONTINUOUS ACTION SPACE

**Action Certificate.** Similar to the discrete-action case, we first prove Theorem C.5, and then prove Theorem 4.7 as a special case of Theorem C.5.

*Proof of Theorem C.5.* To understand the intuition of element-wise median operation in continuous action space, let us first start with an intuitive example: consider 5 arbitrary numbers $x_1, ..., x_5$, if we already know 3 of them $x_1, x_2, x_3$, then it is certain that $\min(x_1, x_2, x_3) \leq \mathsf{Median}(x_1, \cdots, x_5) \leq \max(x_1, x_2, x_3)$. Therefore, when purely benign k-samples form the majority (Condition 4.6), the element-wise median action falls into the range of actions produced by safe messages.

To be more general, in a continuous action space, $\tilde{a} \in \mathsf{Range}(\mathcal{A}_{\text{benign}})$ is equivalent to the condition that out of the $D$ sampled k-samples, purely benign k-samples make up the majority. There are $\sum_{j=\tilde{D}}^{D} \binom{n_2}{j} \binom{n1-n2}{D-j}$ such combinations in terms of the sampling of $D$, where $\tilde{D} = \lfloor \frac{D}{2} \rfloor + 1$. Once again the total number of combinations is $\binom{n_1}{D}$. Therefore, we get Equation (17). $\square$

*Proof of Theorem 4.7.* The proof of Theorem 4.7 follows as a special case of Theorem C.5 when $D = \binom{N-1}{k} = n_1$. In this case, the only non-zero term left in the numerator of $p_D$ is $\binom{n_2}{j}\binom{n_1-n_2}{n_1-j} = \binom{n_2}{n_2}\binom{n_1-n_2}{n_1-n_2} = 1$ (we need $n_2 \geq j$ and $n_1 - n_2 \geq n_1 - j$ to keep the numerator from vanishing, which implies $j = n_2$, which is no lower than $\tilde{D}$ since $n_2 > n_1/2$ due to Condition 4.6). Hence we have $p_D = 1$. $\square$

**Reward Certificate.** Next, we derive the reward guarantee for the continuous-action case.

*Proof of Theorem C.3.* We let $\mathbb{P}(a|s; \pi)$ be the probability of the message-ablation policy $\hat{\pi}$ taking action $a$ at state $s$, where $\pi$ can be either the message-ablation policy $\hat{\pi}$ or the message-ensemble policy $\tilde{\pi}$. Note that this is a conditional probability function, and the policy does not necessarily observe $s$.

Without loss of generality, let $\nu^*$ be the optimal attacking algorithm that minimizes $\tilde{V}_{\nu}^{\tilde{\pi}}$. Let $\mathcal{A}_s$ denote the range of benign action at state $s$ induced by the current message-ablation policy $\hat{\pi}$. Then we have

$$
\sup_{s \in \mathcal{S}} \left| V^{\hat{\pi}}(s) - \tilde{V}_{\nu^*}^{\tilde{\pi}}(s) \right|
$$

$$
= \sup_{s \in \mathcal{S}} \left| \mathbb{E}_{a \sim \mathbb{P}(a|s; \hat{\pi})} \left[ R(s,a) + \gamma \int P(s'|s,a) V^{\hat{\pi}}(s') \mathrm{d}s' \right] - \mathbb{E}_{a \sim \mathbb{P}(a|s; \tilde{\pi})} \left[ R(s,a) + \gamma \int P(s'|s,a) \tilde{V}_{\nu^*}^{\tilde{\pi}}(s') \mathrm{d}s' \right] \right|
$$

$$
\leq \sup_{s \in \mathcal{S}} \sup_{a_1, a_2 \in \mathcal{A}_s} \left| R(s,a_1) + \gamma \int P(s'|s,a_1) V^{\hat{\pi}}(s') \mathrm{d}s' - R(s,a_2) - \gamma \int P(s'|s,a_2) \tilde{V}_{\nu^*}^{\tilde{\pi}}(s)(s') \mathrm{d}s' \right|
$$

$$
\leq \sup_{s \in \mathcal{S}} \sup_{a_1, a_2 \in \mathcal{A}_s} |R(s,a_1) - R(s,a_2)| + \sup_{a_1, a_2 \in \mathcal{A}_s} \left| \gamma \int P(s'|s,a_1) V^{\hat{\pi}}(s') \mathrm{d}s' - \gamma \int P(s'|s,a_2) \tilde{V}_{\nu^*}^{\tilde{\pi}}(s') \mathrm{d}s' \right|
$$

$$
\leq \epsilon_R + \gamma \sup_{s \in \mathcal{S}} \sup_{a_1, a_2 \in \mathcal{A}_s} \left| \int P(s'|s,a_1) V^{\hat{\pi}}(s') \mathrm{d}s' - \int P(s'|s,a_2) \tilde{V}_{\nu^*}^{\tilde{\pi}}(s') \mathrm{d}s' \right|
$$

$$
\leq \epsilon_R + \gamma \sup_{s \in \mathcal{S}} \sup_{a_1, a_2 \in \mathcal{A}_s} \left| \int P(s'|s,a_1) V^{\hat{\pi}}(s') \mathrm{d}s' - \int P(s'|s,a_1) \tilde{V}_{\nu^*}^{\tilde{\pi}}(s') \mathrm{d}s' \right|
$$

$$
+ \gamma \sup_{s \in \mathcal{S}} \sup_{a_1, a_2 \in \mathcal{A}_s} \left| \int P(s'|s,a_1) \tilde{V}_{\nu^*}^{\tilde{\pi}}(s') \mathrm{d}s' - \int P(s'|s,a_2) \tilde{V}_{\nu^*}^{\tilde{\pi}}(s') \mathrm{d}s' \right|
$$

$$
\leq \epsilon_R + \gamma \sup_{s \in \mathcal{S}} \left| V^{\hat{\pi}}(s) - \tilde{V}_{\nu^*}^{\tilde{\pi}}(s) \right| + \gamma \left| \int P(s'|s,a_1) \tilde{V}_{\nu^*}^{\tilde{\pi}}(s') \mathrm{d}s' - \int P(s'|s,a_2) \tilde{V}_{\nu^*}^{\tilde{\pi}}(s') \mathrm{d}s' \right|
$$

$$
\leq \epsilon_R + \gamma \sup_{s \in \mathcal{S}} \left| V^{\hat{\pi}}(s) - \tilde{V}_{\nu^*}^{\tilde{\pi}}(s) \right| + \gamma V_{\max} \epsilon_P.
$$

$$
\text{(18)}
$$

By solving for the recurrence relation over $\sup_{s\in\mathcal{S}}\left|V^{\hat{\pi}}(s)-\tilde{V}_{\nu^*}^{\tilde{\pi}}(s)\right|$, we obtain

$$\sup_{s\in\mathcal{S}}\left|V^{\hat{\pi}}(s)-\tilde{V}_{\nu^*}^{\tilde{\pi}}(s)\right|\leq\frac{\epsilon_R+\gamma V_{\max}\epsilon_P}{1-\gamma}. \tag{19}$$

which leads to the desired relation in Theorem C.3.

$\square$

## E  EXPERIMENT DETAILS AND ADDITIONAL RESULTS

### E.1  ENVIRONMENT DESCRIPTION

#### E.1.1  FOODCOLLECTOR

The FoodCollector environment is a 2D particle world adapted from the WaterWorld task in PettingZoo (Terry et al., 2020), shown by Figure 4(left). There are $N = 9$ agents with different colors, and $N$ foods with colors corresponding to the $N$ agents. Agents are rewarded when eating foods with the same color. A big round obstacle is located in the center of the map, which the agent cannot go through. There are some poisons (shown as black dots) in the environment, and the agents get penalized whenever they touch the poison. Each agent has 6 sensors that detect the objects around it, including the poisons and the colored foods. The game is episodic, with horizon set to be 200. In the beginning of each episode, the agents, foods and poisons are randomly generated in the world.

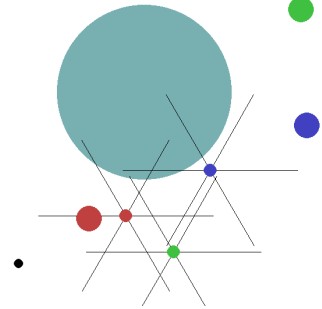

**Figure 4:** The FoodCollector Environment. For figure readability, we only show 3 agents colored as red/green/blue. In our experiments, there are 9 agents.

**State Observation**  Each of the 6 sensors can detect the following values when the corresponding element is within the sensor's range (the corresponding dimensions are 0's if nothing is detected):
(1) (if detects a food) the distance to a food (real-valued);
(2) (if detects a food) the color of the food (one-hot);
(3) (if detects an obstacle) the distance to the obstacle (real-valued);
(4) (if detects the boundary) the distance to the boundary (real-valued);
(5) (if detects another agent) the distance to another agent (real-valued).
The observation of an agent includes an agent-identifier (one-hot encoding), its own location (2D coordinates), its own velocity, two flags of colliding with its food and colliding with poison, and the above sensory inputs. Therefore, the observation space is a $(7N + 30)$-dimensional vector space.

**Agent Action**  The action space can be either *discrete* or *continuous*. For the discrete version, there are 9 actions including 8 moving directions (north, northwest, west, southwest, south, southeast, east, northeast) and 1 no-move action. For the continuous version, the action is an acceleration decision, denoted by a 2-dimensional real-valued vector, with each coordinate taking values in $[-0.01, 0.01]$.

**Reward Function**  At every step, each agent will receives a negative reward $-0.5$ if it has not eaten all its food. In addition, it receives extra $-1$ reward if it collides with a poison. Therefore, every agent is expected to explore the environment and eat all food as fast as possible. The team reward is calculated by the average of all agents' local rewards. Note that the agents' actions do not affect each other, because they have different target foods to collect. *Agents collaborate only via communication introduced below.*

**Communication Protocol**  Due to the limited sensory range, every agent can only see the objects around it and thus only partially observes the world. Therefore, communication among agents can help them find their foods much faster. Since our focus is to defend against adversarially per-

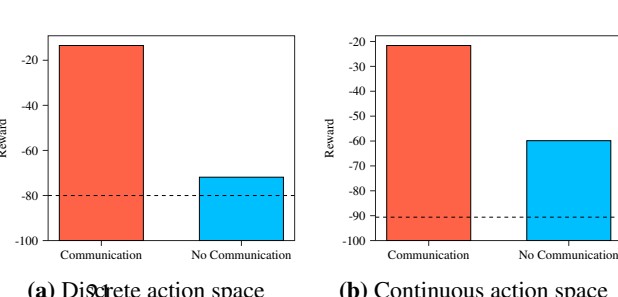

**(a)** Discrete action space

**(b)** Continuous action space

**Figure 5: FoodCollector**: Reward of agents trained by PPO with communication v.s. without communication. Black dashed line

turbed communications, we first define a valid and beneficial communication protocol, where an agent sends a message to a receiver once it observes a food with the receiver's color. For example, if a red agent encounters a blue food, it can then send a message to the blue agent so that the blue agent knows where to find its food. To remember the up-to-date communication, every agent maintains a list of most recent $N-1$ messages sent from other $N-1$ agents. A message contains the sender's current location and the relative distance to the food (recorded by the 6 sensors), which are bounded between -1 and 1. Therefore, a message is a 8-dimensional vector, and each agent's communication list has $8(N-1)$ dimensions in total.

**Communication Gain**  During training with communication, we concatenate the observation and the communication list of the agent to an MLP-based policy, compared to the non-communicative case where the policy only takes in local observations. More implementation details are in Appendix E.2.1. As verified in Figure 5, communication does help the agent to obtain a much higher reward in both discrete action and continuous action cases, which suggests that the agents tend to rely heavily on the communication messages for finding their food.

### E.1.2  INVENTORYMANAGER

The *InventoryManager* environment is an inventory management setup, where $N = 10$ cooperative heterogeneous distributors carry inventory for $M = 3$ products. A population of $B = 300$ buyers request a product from a randomly selected distributor agent according to a demand distribution $\mathbf{p} = [p_1, \ldots, p_M]$. We denote the demand realization for distributor $i$ with $\mathbf{d_i} = [d_{i,1}, \ldots, d_{i,M}]$. Distributor agents manage their inventory by restocking products through interacting with the buyers. The game is episodic with horizon set to $50$. At the beginning of each episode, a realization of the demand distribution $\mathbf{p}$ is randomly generated and the distributors' inventory for each product is randomly initialized from $[0, \frac{B}{N}]$, where $\frac{B}{N}$ is the expected number of buyers per distributor. The distributor agents are penalized for mismatch between their inventory and the demand for a product, and they aim to restock enough units of a product at each step to prevent insufficient inventory without accruing a surplus at the end of each step.

**State Observation**  A distributor agent's observation includes its inventory for each product, and the products that were requested by buyers during the previous step. The observation space is a $2M$-dimensional vector.

**Agent Action**  Distributors manage their inventory by restocking new units of each product or discarding part of the leftover inventory at the beginning of each step. Hence, agents take both positive and negative actions denoted by an $M$-dimensional vector, and the action space can be either discrete or continuous. In our experiments, we use continuous actions assuming that products are divisible and distributors can restock and hold fractions of a product unit.

**Reward Function**  During each step, agent $i$'s reward is defined as $r_i = -|| \max(\mathbf{I}_i + \mathbf{a}_i, 0) - \mathbf{d}_i ||_2$, where $\mathbf{I}_i$ denotes the agents initial inventory vector, and $\mathbf{a}_i$ denotes the inventory restock vector from action policy $\pi_i$. Note that the agents' actions do not affect each other. *Agents collaborate only via communication introduced below.*

**Communication Protocol**  Distributors learn the demand distribution and optimize their inventory by interacting with their own customers (i.e., portion of buyers that request a product from that distributor). Distributors would benefit from sharing their observed demands with each other, so that they could estimate the demand distribution more accurately for managing inventory. At the end of each step, a distributor communicates an $M$-dimensional vector reporting its observed demands to all other agents. In the case of adversarial communications, this message may differ from the agents' truly observed demands.

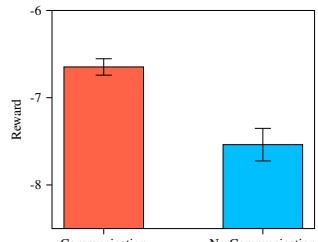

**Figure 6: InventoryManager**: Reward of agents trained with communication v.s. without communication.

**Communication Gain** During training with communication, messages received from all other agents are concatenated to the agent's observation, which is used to train an MLP-based policy, compared to the non-communicative case where the policy is trained using only local observations. As observed from Figure 6, communication helps agents obtain higher rewards since they are able to manage their inventory based on the overall demands observed across the population of buyers rather than their local observation. Results are reported by averaging rewards corresponding to 5 experiments run with different training seeds.

### E.1.3 MARL-MNIST

We use the environment setup proposed by Mousavi et al. Mousavi et al. (2019), where agents collaboratively classify an unknown image by their observations and inter-agent communication. More specifically, we use $N = 9$ agents in the MNIST dataset of handwritten digits (LeCun et al., 1998). The dataset consists of 60,000 training images and 10,000 testing images, where each image has $28 \times 28$ pixels. There are $h = 5$ steps in an episode. In the beginning, all agents start from a pre-determined spatial configuration. At every step, each agent observes a local $5 \times 5$ patch, performs some local data processing, and shares the result with neighboring agents (we use a fully connected communication graph). With given observation and communication, each agent outputs an action in $\{Up, Down, Left, Right\}$. By each movement, the agent is translated in the desired direction by 5 pixels. In the end of an episode, agents make predictions, and all of them are rewarded by $-$prediction_loss.

### E.1.4 TRAFFIC JUNCTION

We use the same environment setup as the one in (Singh et al., 2019). In this environment, $N$ cars enter a junction from all entry points with a given probability $p_{arr}$. The maximum number of cars at any given time in the junction is limited. Cars can take two actions at each time-step, *gas* and *brake* respectively. At every step, each agent observes its previous action, route identifier, and a vector specifying the sum of one-hot vectors for all objects present at that agent's location. A collision occurs when two cars are at the same location, and the agent will get a -10 reward if a collision occurs. In addition, to prevent traffic jams, for each time step when the agent is in the junction, it will get a negative reward of -0.01. The horizon of the environment is set to be 60. We use the hardest environment setup as in (Singh et al., 2019), where we have 10 cars, the vision is set to zero, and the horizon of the environment is set to 80.

### E.2 IMPLEMENTATION DETAILS

### E.2.1 FOODCOLLECTOR

**Implementation of Trainer** In our experiments, we use the Proximal Policy Optimization (PPO) (Schulman et al., 2017) algorithm to train all agents (with parameter sharing among agents) as well as the attackers. Specifically, we adapt from the elegant OpenAI Spinning Up (Achiam, 2018) Implementation for PPO training algorithm. On top of the Spinning Up PPO implementation, we also keep track of the running average and standard deviation of the observation and normalize the observation. All experiments are conducted on NVIDIA GeForce RTX 2080 Ti GPUs.

For the policy network, we use a multi-layer perceptron (MLP) with two hidden layers of size 64. For a discrete action space, a categorical output distribution is used. For a continuous action space, since the valid action is bounded within a small range [-0.01,0.01], we parameterize the policy as a Beta distribution, which has been proposed in previous works to better solve reinforcement learning problems with bounded actions (Chou et al., 2017). In particular, we parameterize the Beta distribution by $\alpha_\theta$ and $\beta_\theta$, such that $\alpha = \log(1 + e^{\alpha_\theta(s)}) + 1$ and $\beta = \log(1 + e^{\beta_\theta(s)}) + 1$ (1 is added to make sure that $\alpha, \beta \geq 1$). Then, $\pi(a|s) = f(\frac{a-h}{2h}; \alpha, \beta)$, where $h = 0.01$, and $f(x; \alpha, \beta) = \frac{\Gamma(\alpha+\beta)}{\Gamma(\alpha)\Gamma(\beta)} x^{\alpha-1}(1-x)^{\beta-1}$ is the density function of the Beta Distribution. For the value network, we also use an MLP with two hidden layers of size 64.

In terms of other hyperparameters used in the experiments, we use a learning rate of 0.0003 for the policy network, and a learning rate of 0.001 is used for the value network. We use the Adam

optimizer with $\beta_1 = 0.99$ and $\beta_2 = 0.999$. For every training epoch, the PPO agent interacts with the environment for 4000 steps, and it is trained for 500 epochs in our experiments.

**Attacker** An attacker maps its own observation to the malicious communication messages that it will send to the victim agent. Thus, the action space of the attacker is the communication space of a benign agent, which is bounded between -1 and 1.

- *Heuristic Attacker* We implement a fast and naive attacking method for the adversary. At every dimension, the naive attacker randomly picks 1 or -1 as its action, and then sends the perturbed message which consists entirely of 1 or -1 to the victim agent.
- *Adaptive RL Attacker* We use the PPO algorithm to train the attacker, where we set the reward of the attacker to be the negative reward of the victim. The attacker uses a Gaussian policy, where the action is clipped to be in the valid communication range. The network architecture and all other hyperparameter settings follow the exact same from the clean agent training.

**Implementation of Baselines**

- *Vanilla Learning* For Vanilla method, we train a shared policy network to map observations and communication to actions.
- *Adversarial Training (AT)* For adversarial training, we alternate between the training of attacker and the training of the victim agent. Both the victim and attackers are trained by PPO. For every 200 training epochs, we switch the training, where we either fix the trained victim and train the attacker for the victim or fix the trained attacker and train the victim under attack. We continue this process for 10 iterations.

Note that the messages are symmetric (of the same format), we shuffle the messages before feeding them into the policy network for both Vanilla and AT, to reduce the bias caused by agent order. We find that shuffling the messages helps the agent converge much faster (50% fewer total steps). Note that AME randomly selects k-samples and thus messages are also shuffled.

### E.2.2 INVENTORYMANAGER

**Implementation of Trainer** As in the FoodCollector experiments, we use the PPO algorithm to train all agent action policy as well as adversarial agent communication policies. We use the same MLP-based policy and value networks as the FoodCollector but parameterize the policy as a Gaussian distribution. The PPO agent interacts with the environment for 50 steps, and it is trained for 10000 episodes. The learning rate is set to be 0.0003 for the policy network, and 0.001 for the value network.

**Attacker** The attacker uses its observations to communicate malicious messages to victim distributors, and its action space is the communication space of a benign agent. We consider the following non-adaptive and adaptive attackers:

- *Heuristic Attackers*: The attacker's goal is to harm the victim agent by misreporting its observed demands so that the victim distributor under-estimates or over-estimates the re-stocking of products. In our experiments, we evaluate the effectiveness of defense strategies against the following attack strategies:
    - Perm-Attack: The communication message is a random permutation of the true demand vector observed by the attacker.
    - Swap-Attack: In order to construct a communication vector as different as possible from its observed demand, the attacker reports the most requested products as the least request ones and vice versa. Therefore, the highest demand among the products is interchanged with the lowest demand, the second highest demand is interchanged with the second lowest demand and so forth.
    - Flip-Attack: Adversary $i$ modifies its observed demand $\mathbf{d}_i$ by mirroring it with respect to $\eta = \frac{1}{M} \sum_{j=1}^{M} d_{ij}$, such that products demanded less than $\eta$ are reported as being requested more, and conversely, highly demanded products are reported as less popular.
- *Adaptive Attacker*: The attacker communication policy is trained using the PPO algorithm, and its reward is set as the negative reward of the victim agent. The attacker uses a

Gaussian policy with a softmax activation in the output layer to learn a adversarial probability distribution across products, which is then scaled by the total observed demands $\sum_{j=1}^{M} d_{ij}$ to construct the communication message.

**Implementation of Baselines**

- *Vanilla Learning*: In the vanilla training method with no defense mechanism against adversarial communication, we train a shared policy network using agents' local observations and their received communication messages.

- *Adversarial Training (AT)*: We alternate between training the agent action policy and the victim communication policy, both using PPO. For the first iteration we train the policies for 10000 episodes, and then use 1000 episodes for 5 additional training alteration iterations for both the action policy and the adversarial communication policy. For more efficient adversarial training, we first shuffle the received communication messages before feeding them into the policy network. Consequently, the trained policy treats communications received from different agents in a similar manner, and we are able to only train the policy with a fixed set of adversarial agents rather than training the network on all possible combinations of adversarial agents.

For Vanilla and AT, we do not shuffle the communication messages being input to the policy, as we did not observe improved convergence, as in the FoodCollector environment.

### E.2.3   MARL-MNIST

We use the same network architecture and hyperparameter setting as Mousavi et al. Mousavi et al. (2019), which are implemented in Berrien (2019).

**Network Architecture**   Concretely, at every step, the input of every agent contains 3 components: (1) an encoded observation with two convolutional layers followed by vectorization and a fully connected layer; (2) the average of all communication messages from other agents; (3) a position encoding computed by feeding the current position into a single linear layer. These 3 components are concatenated and passed to two independent LSTM modules, one is for an acting policy, another is for a message generator. In the end of an episode, every agent uses its final cell state to generate a prediction using a 2-layer MLP. Then we take the average of the output logits of all agents, and use a softmax function to obtain the final probabilistic label prediction. The reward is the opposite number of the L2 difference between the prediction and the one-hot encoding of true image label.

**Hyperparameters**   In our experiments, we follow the default hyperparameter setting in Berrien (2019). We use $N = 9$ agents. The size of LSTM belief state is 128. The hidden layers have size 160. The message size is set to be 32. The state encoding has size 8. We use an Adam optimizer with learning rate 1e-3. We train the agents in the MNIST dataset for 40 epochs.

**Attackers**   Since the communication messages are learned by neural networks, we perturb the $C$ messages received by each agent. To make sure that the messages are not obviously detectable, we clip every dimension of the perturbed message into the range of $[-3, 3]$. The non-adaptive attacker randomly generates a new message. The adaptive attacker learns a new message generator based on its own belief state, which is trained with learning rate 1e-3 for 50 epochs.

### E.2.4   TRAFFIC JUNCTION

To learn the agents' policy and communication protocol on traffic junctions, we follow exactly the implementation of (Singh et al., 2019).

**Network Architecture**   In particular, the policy of the $j$-th agent in IC3Net takes the form of:

$$g_j^{t+1} = f^g(h_j^t)$$
$$h_j^{t+1}, s_j^{t+1} = LSTM(e(o_j^t) + c_j^t, h_j^t, s_j^t)$$
$$c_j^{t+1} = \frac{1}{N-1} C \sum_{j' \neq j} h_{j'}^{t+1} \odot g_{j'}^{t+1}$$
$$a_j^t = \pi(h_j^t)$$

Here, $g_j^t$ is a binary action that specifies whether the agent $j$ should communicate with other agents, and so it acts as a gating function when calculating the communication messages $c_j^{t+1}$. $f_g(\cdot)$ is a simple network containing a soft-max layer for 2 actions (communicate or not) on top of a linear layer with non-linearity. $e$ is an encoder of the agent's observation and $C$ is a linear transformation matrix for transforming gated average hidden state to a communication tensor that has the same size as the observation encoding.

**Hyperparameter**   The encoder $e$ is a linear layer of size 128, and $f^g(\cdot)$ is also a linear layer of size 128. The LSTM module is of two layers with hidden size 128, and the policy head $\pi$ is a linear layer with an output size 2 equal to the number of actions.

**Attacker**   We perturb the $C$ messages received by each agent. To make sure that the messages are not obviously detectable, we clip every dimension of the perturbed message into the range of $[-0.5, 0.5]$. The non-adaptive attacker randomly generates a new message containing either -0.5 or 0.5. For the adaptive attacker, it is trained with a recurrent PPO through stable-baselines3 (Raffin et al., 2021) for 500,000 steps.

### E.3   ADDITIONAL RESULTS

#### E.3.1   FULL RESULTS OF FOODCOLLECTOR

Figure 7 below shows the results of both discrete-action FoodCollector and continuous-action FoodCollector.

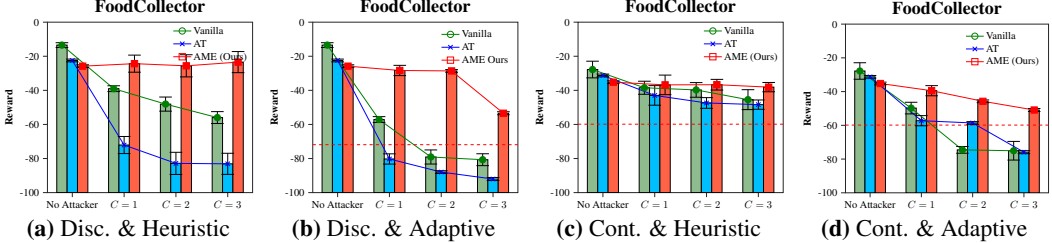

(a) Disc. & Heuristic    (b) Disc. & Adaptive    (c) Cont. & Heuristic    (d) Cont. & Adaptive

**Figure 7:** Rewards of our AME and baselines in FoodCollector, under no attacker and varying numbers of adversaries for adaptive and heuristic (random message) attacks.

**Additional Results on Discrete-action Food-Collector with QMIX**   AME is a generic defense approach that can be used for any RL/MARL learning algorithm. In Figure 8, we show the results of AME combined with an MARL algorithm QMIX (Rashid et al., 2018) in the discrete-action FoodCollector environment (QMIX does not work for continuous actions so it is not applicable in continuous-action Food-Collector and InventoryManager). Compared to the vanilla QMIX algorithm, QMIX+AME achieves much higher robustness and stable performance under various numbers of adversaries.

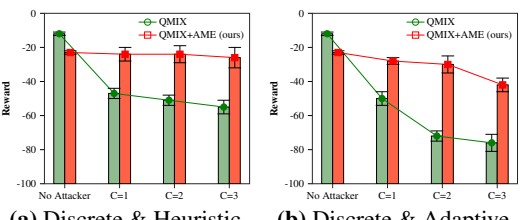

(a) Discrete & Heuristic    (b) Discrete & Adaptive

**Figure 8:** Reward comparison between original QMIX and QMIX combined with our AME in FoodCollector with discrete action space, under no attacker or non-adaptive/adaptive attacks under varying numbers of adversaries. For AME, the ablation size $k$ is set as 2.

#### E.3.2   FULL RESULTS OF INVENTORYMANAGER

Figure 9 below shows the results of InventoryManager under adaptive attacks and several heuristic attacks.

#### E.3.3   FULL RESULTS OF MARL-MNIST

Figure 10 demonstrates the robust performance of the MARL algorithm proposed by Mousavi et al. (2019) with our AME defense or baseline defenses (Vanilla and AT). We set $N = 9$ and $k = 2$ for all experiments. Under learned adaptive attackers, the original MARL classifier (Vanilla) (Mousavi

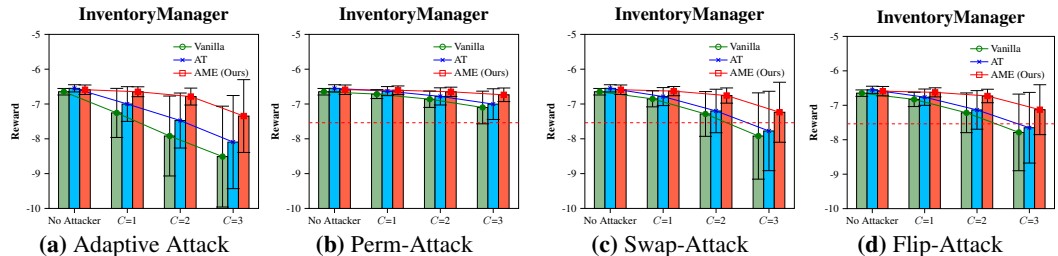

**Figure 9:** Rewards of our AME and baselines in InventoryManager, under no attacker and varying numbers of adversaries for adaptive and various heuristic attacks.

et al., 2019) without AME suffers from significant performance drop in terms of both precision and recall. Defending with adversarial training (AT) does not achieve good robustness, either. But AME considerably improves the robustness of agents across different numbers of attackers.

Under random attacks, we find that the original MARL classifier (Mousavi et al., 2019) is moderately robust when a few communication signals are randomly perturbed. However, when noise exists in many communication channels (e.g. $C = 6$), the performance decreases a lot. In contrast, our AME still achieves high performance when $C = 6$ communication messages are corrupted, even if the guarantee of ablation size $k$ only holds for $C \leq 2$. Therefore, we again emphasize the the theoretical guarantee considers the worst-case strong attack, while under a relatively weak attack, we can achieve better robustness beyond what the theory suggests.

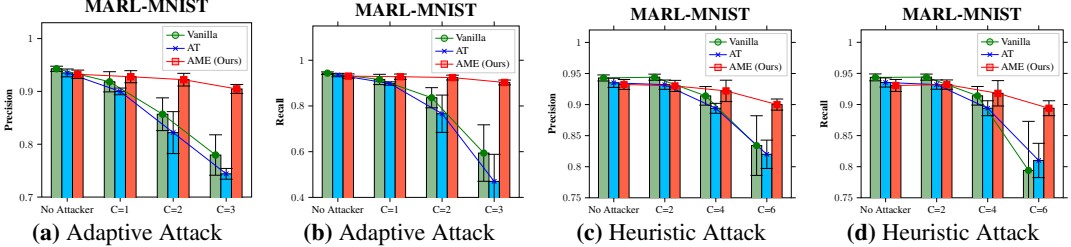

**Figure 10:** (**MARL-MNIST**): Precision and recall of MARL classification on MNIST without or with AME, under learned adaptive attacks and non-adaptive random attacks. All results are averaged over 5 random seeds.

### E.3.4 ADDITIONAL RESULTS OF HYPERPARAMETER TEST

In addition to Figure 3, we also provide the plot for hyper-parameter tests in discrete-action FoodCollector and InventoryManager in Figure 11 below.

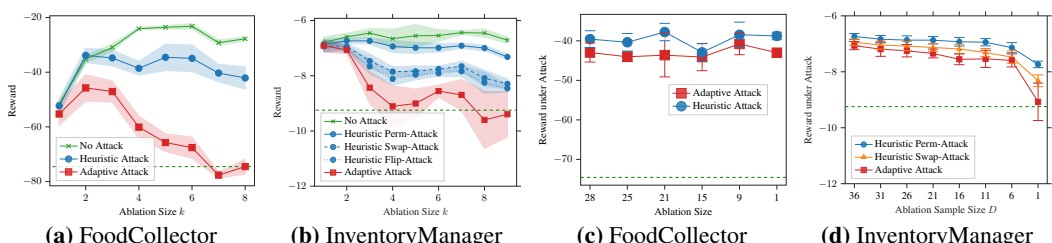

**Figure 11:** Hyper-parameter tests of ablation size $k$ and sample size $D$. We show how natural reward and attacked performance change with (**a**) various $k$ in continuous-action FoodCollector, (**b**) various $D$ in continuous-action FoodCollector, (**c**) various $k$ in InventoryManager, and (**d**) various $D$ in InventoryManager. Dashed green lines refer to the performance of Vanilla agent under $C = 2$ attacks.

### E.3.5 INTERPRETATION OF ADVERSARIAL TRAINING

As we observed in Figure 2, adversarial training (AT) does not improve the robustness of the agent in both discrete and continuous action space, although AT achieves good robustness againt $\ell_p$ attacks in vision tasks (Madry et al., 2018; Zhang et al., 2019), and against observation attacks (Zhang et al., 2021; Sun et al., 2022) and action attacks (Pinto et al., 2017) in RL. We hypothesize that it is due to (1) the large number of total agents, (2) the uncertainty of adversarial message channels, and (3) the relatively large perturbation length. To be more specific, in related works (Pinto et al., 2017; Zhang et al., 2021; Sun et al., 2022), an agent and an attacker are alternately trained, during which the agent learns to adapt to the learned attacker. However, in the threat model we consider, there are $C$ out of $N$ messages significantly perturbed, which it is hard for the agent to adapt to attacks during alternate training.

## F ADDITIONAL DISCUSSION ON THE CHOICE OF ABLATION SIZE $k$

Our AME defensive mechanism proposed in Section 4.1 requires an extra hyperparameter: the ablation size $k$. (The sample size $D$ is only for the partial-sample variant introduced in Section 4.3, not needed in the original form of AME.) In this section, we discuss the selection of $k$ in practice. We start by discussing the relationship between $k$ and the required theoretical conditions.

**I. (For Continuous Action Space) Condition 4.6 and $k$**

We first decompose Equation (6) as below.

$$\binom{N-1-C}{k} > \frac{1}{2}\binom{N-1}{k} \tag{20}$$

$$\frac{(N-1-C)!}{(N-1-C-k)!k!} > \frac{(N-1)!}{2(N-1-k)!k!} \tag{21}$$

$$1 > \frac{(N-1)\cdots(N-C)}{2(N-k-1)\cdots(N-k-C)} \tag{22}$$

Therefore, Equation (6) is equivalent to

$$(N-k-1)\cdots(N-k-C) > \frac{1}{2}(N-1)\cdots(N-C) \tag{23}$$

When $k = 1$, the above inequality becomes

$$(N-2)\cdots(N-C)(N-C-1) > \frac{1}{2}(N-1)(N-2)\cdots(N-C) \tag{24}$$

$$(N-C-1) > \frac{1}{2}(N-1) \tag{25}$$

$$\frac{1}{2}(N-1) > C, \tag{26}$$

which holds under Assumption 3.1. Therefore, **$k = 1$ is always a feasible solution for Condition 4.6.**

Then, when $C$ is fixed and $k$ goes up from 1, the LHS of Equation (23) goes down while the RHS does not change. Therefore, for a given number of agents ($N$) and a fixed number of adversaries ($C$), there exists an integer $k_0$ such that any $k \leq k_0$ satisfies Condition 4.6. In practice, if we have an estimate of the number of adversarial messages that we would like to defend against, then we could choose the maximum $k$ satisfying Equation (6).

On the other hand, when $k$ is fixed, there exists a $C_0$ such that any $C \leq C_0$ can let Equation (23) hold. Therefore, if $C$ is unknown, for any selection of $k$, Equation (23) can justify the maximum number of adversaries for the current selection.

From Equation (23), we can see the interdependence between three parameters: total number of agents $N$, number of adversaries $C$, and the ablation size $k$. In Figure 12 we visualize their relationship by fixing one variable at a time. From these figures, we can see an obvious trade-off between $C$ and $k$ — if there are more adversaries, $k$ has to be set smaller. But when $C$ is small, e.g. $C = 1$, $k$ can be relatively large, so the agent does not need to compromise much natural performance to achieve robustness.

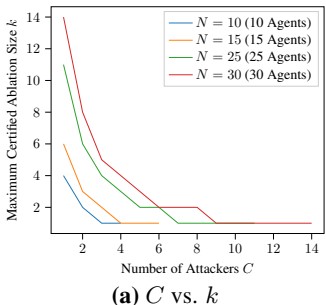 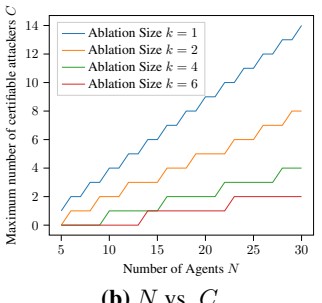 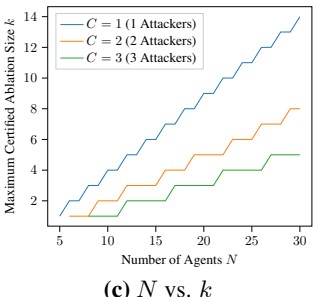

**(a)** $C$ vs. $k$      **(b)** $N$ vs. $C$      **(c)** $N$ vs. $k$

**Figure 12:** Relationship between the total number of agents $N$, number of attackers $C$, and ablation size $k$. **(a)**: Maximum certifiable ablation size $k$ under different number of attackers. **(b)**: Maximum defensible number of attackers $C$ with different total number of agents $N$. **(c)**: Maximum certifiable ablation size $k$ with different total number of agents $N$.

## II. (For Discrete Action Space) Condition 4.4 and $k$

Different from Condition 4.6 which is independent of the environment, Condition 4.4 required in a discrete action space is related to the environment and the communication quality. Intuitively, Condition 4.4 can be satisfied if the benign messages can reach some "consensus", i.e., there are enough purely benign k-samples voting for the same action. This can be achieved when the environment is relatively deterministic (e.g., there is a certainly optimal direction to go). Fortunately, Condition 4.4 can be checked during training time, and we can train the message-ablation policy and the communication policy to increase $u_{\max}$ as much as possible.

On the other hand, Condition 4.4 is also related to the selection of ablation size $k$. The ratio of contaminated votes is $\frac{\binom{N-1}{k} - \binom{N-1-C}{k}}{\binom{N-1}{k}}$, and it is easy to show that this rate increases as $k$ when $k \leq N - C - 1$. Therefore, when $k$ is smaller, it is relatively easier to satisfy Condition 4.4 as the adversarial messages can take over a smaller proportion of the total number of votes.

***$k$ Balances between Natural Performance and Robustness*** The above analysis shows that a smaller $k$ makes the agent more robust, while the natural performance may be sacrificed as the message-ablation policy makes decisions based on less benign information. Such a trade-off between natural performance and robustness is common in the literature of adversarial learning (Tsipras et al., 2019; Zhang et al., 2019). Note that AME randomize the message ablation process at every step, such that important messages are less likely to be missed, as communication from every agent usually does not drastically change over time. Our experimental results in Section 6 show that the robustness gain of AME is much greater than the compromise in natural performance in all tested domains.

**How to Select $k$ in Practice?** In practice, we suggest setting $k$ to be the largest integer satisfying Equation (6). If higher robustness is needed, then $k$ can be further decreased. If robustness is not the major concern while higher natural performance is required, one can increase $k$. Note that if $k = N - 1$, AME degenerates to the original vanilla policy without defense.

**What If Conditions Are Not Satisfied?** Even if Condition 4.6 and Condition 4.4 are not satisfied, the agent can still be robust under attacks as verified in our experiments (AME with $k = 2$ still achieves relatively robust performance under $C = 3$ which exceeds the theoretically largest number of certifiable adversaries). Because these conditions are needed for the certificates which consider the theoretically worst-case attacks. However, in practice, an attacker has restricted power and knowledge (e.g., it does not know the victim policy/reward, and does not know the environment dynamics as prior), and is likely to be even weaker than the learned adaptive white-box attacker we use in experiments. As a result, even if a larger $k$ may break the conditions, it can still improve the empirical robustness of an agent in practice. Our Figure 3 and Figure 11, where AME maintains good empirical performance for almost all choices of $k < N - 1$. ($K = N - 1$ is the original vanilla policy without defense.)

**Extension: Adaptive Defense with Different $k$'s** Moreover, to allow higher flexibility, one can train multiple message-ablation policies with different selections of $k$'s during training. Then, an

adaptive strategy can be used in test time. For example, if $u_{\max}$ is too small, we can use a larger $k$ with the corresponding trained message-ablation policy.

**Extension: Gaining both Natural Performance and Robustness by Attack Detection** From the analysis of the relation between $N$, $C$ and $k$, we can see that when the number of adversaries is large, the corresponding ablation size $k$ is supposed to be smaller. This is reasonable because a more conservative defense is needed against a stronger attacker. But if we can identify some adversarial messages and rule them out before message ablation and ensemble, we can still defend with guarantees using a relatively large $k$. For example, if we have identified $c$ adversarial messages, then we only need to deal with the remaining $C - c$ adversarial messages out of $N - 1 - c$ messages. By Equation (6), a larger $k$ can be used compared to defending $C$ adversarial messages out of $N - 1$ messages. We also provide an adversary detection algorithm in Appendix G using a similar idea of AME.

# G  DISCUSSION: DETECTING MALICIOUS MESSAGES WITH ABLATION

As discussed in Appendix F, to defend against a larger $C$, one has to choose a relatively small $k$ for certifiable performance. However, Figure 11 suggests that a small $k$ also sacrifices the natural performance of the agent to obtain higher robustness. This is known as the trade-off between robustness and accuracy (Tsipras et al., 2019; Zhang et al., 2019). Can we achieve better robustness while not sacrificing much natural performance, or obtain higher natural reward while not losing robustness?

We point out that with our proposed AME defense, it is possible to choose a larger $k$ than what is required by Condition 4.6 without sacrificing robust performance by identifying the malicious messages beforehand. The idea is to detect the adversarial messages and to rule out them before message ablation and ensemble, during the test time.

We hypothesize that given a well-trained victim policy, malicious messages tend to mislead the victim agent to take an action that is "far away" from a "good" action that the victim is supposed to take. To verify this hypothesis, we first train a message-ablation policy $\hat{\pi}_i$ with $k = 1$ for agent $i$. Then for every communication message that another agent $j$ sends, we compute the action $a_j$ that the victim policy $\hat{\pi}_i$ chooses based on all message subsets containing message $m_{j \to i}$, i.e. $a_j = \hat{\pi}(\tau_i, m_{j \to i})$ (note that $k = 1$ so $\hat{\pi}$ only takes in one message at a time). We then define the *action bias*

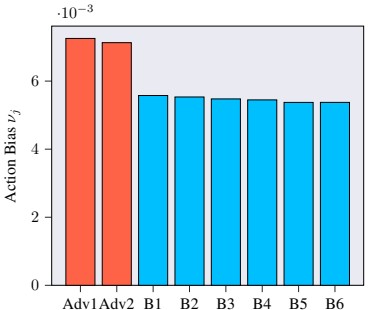

**Figure 13:** Attacker identification based on action bias $\nu_j$. Adv1 and Adv2 stands for two agents hacked by the attacker. B1 up to B6 stands for six other benign agents.

as $\beta_j = \|a_j - \mathsf{Median}\{a_k\}_{k=1}^N\|_1$. Based on our hypothesis, an agent which has been hacked by attackers should induce a significantly larger action bias since they are trying to mislead the victim to take a completely different action. Here, we execute the policy of two hacked agents together with six other good agents for twenty episodes and calculate the average action bias for each agent. As shown in in Figure 13, the agents hacked by attackers indeed induce a larger action bias compared to other benign agents, which suggests the effectiveness of identifying the malicious messages by action bias.

After filtering out $c$ messages, one could compute the required $k$ by Equation equation 6 based on $C - c$ malicious messages and $N - 1 - c$ total messages, which can be larger than the largest $k$ induced by $C$ malicious messages out of $N - 1$ total messages. For example, if $N = 30$, $C = 3$, then the largest $k$ satisfying Equation equation 6 is 5, but when 1 adversarial message is filtered out, the largest $k$ that can defend against the remaining 2 adversarial messages is 8. Although the adversary identification is not theoretically guaranteed to be accurate, Figure 13 demonstrates the effectiveness of the adversarial message detection, which, combined with AME with larger $k$'s, has the potential to achieve high natural performance and strong robustness in practice.

