# OpenReview forum: "Certifiably Robust Policy Learning against Adversarial Multi-Agent Communication"
_ICLR.cc/2023/Conference — ICLR 2023 poster_

### Official Review · Reviewer_iqJ1 · 2022-10-25

**Confidence:** 4
**Correctness:** 4
**Technical Novelty And Significance:** 3
**Empirical Novelty And Significance:** 3
**Recommendation:** 8

**Clarity, Quality, Novelty And Reproducibility:**

**Clarity and Presentation**
Overall the paper is clearly written and can easily be followed.
I think the presentation could be improved by including at least Algorithm 2 in the main paper.

**Reproducibility**
As the AME approach is relatively simple, it should be straightforward to re-implement. However, to recover the exact experiments, I encourage the authors to provide their code to the eventual readers.

**Further Questions**
Some small questions, that remained after reading:
- I assume that in the evaluation all agents employ AME. Is this the case? It would be potentially interesting to see settings where an AME agent interacts with Vanilla ones. Can you comment on this setting?
- You discuss the importance of redundancy in the observation for AME to work well. I could imagine that there are cases in which some information is only observed by one (or very few) agent(s). Can you further quantify or discuss this aspect?

**Minor Formatting/Typos**
- The citation style after MARL-MNIST and Traffic Junction is inconsistent (page 8).





**Strength And Weaknesses:**

**Strengths:**
- Strong and relevant attacker model.
- Simple, well-motivated certifiable defense.
- Mathematical details appear to be sound.
- Thorough evaluation.

**Weaknesses:**
- The paper does not discuss the robustness certificate in depth (although the appendix does). Generally, I feel, the paper might benefit from including more from appendices B and C in the main paper (at the cost of shortening some other discussion).
- While the evaluation is thorough, Section 6 was hard to follow (as it required a lot of going back and forth between multiple paragraphs and Figure 3).



**Summary Of The Paper:**

This paper considers adversarial robustness in the communicative multi-agent reinforcement learning setting (CMARL), where $N$ agents communicate with each other.
At each step, an agent observes (their part) of the environment, receives messages from the other $N-1$ agents, and proposes an action.
The thread model considered in this paper assumes that for an (arbitrary) individual agent up to $C \leq \frac{N-1}{2}$ of the received messages can be arbitrarily changed. This setting is formalized and a certifiable defense, AME, for it is proposed.

AME utilizes a random ablation approach, as proposed in [1]: the action is determined as the aggregate over the output of an arbitrary underlying policy evaluated on $D$ randomly sampled subsets of the received messages.
Each subset is of size $k \leq N-1$, where the parameter $k$ controls a trade-off between high robustness ($k$ low) and high performance ($k=N-1$; the standard underlying algorithm). The aggregation operation depends on whether continuous or discrete actions are considered.

AME is evaluated in different CMARL environments, where AME (usually) reduces the performance when no adversary is present.
However, in case an adversary is present AME indeed outperforms standard and adversarial trianing.

[1] Robustness certificates for sparse adversarial attacks by randomized ablation, Alexander Levine and Soheil Feizi, AAAI20


**Summary Of The Review:**

The paper studies an important problem -- communication robustness in CMARL -- and proposes a simple yet effective and well-founded defense. The paper is (largely) well-written and easy to follow.

I thus recommend acceptance.

---

> ### Author Response · Authors · 2022-11-16
> **Response to Reviewer iqJ1 - Part II**
>
> > Q6: You discuss the importance of redundancy in the observation for AME to work well. I could imagine that there are cases in which some information is only observed by one (or very few) agent(s). Can you further quantify or discuss this aspect?
>
> This is a very nice point. We discuss it from the following 2 aspects.
>
> **1. Communication with less information redundancy is intrinsically more vulnerable to attacks.**
>
> We point out that there exists a trade-off between communication efficiency and policy robustness. It is possible that sometimes an agent only needs one (or few) message to make a good decision, resulting in efficient communication. However, if the decision is purely based on a single message, there is no hope to perform well if there is a strong adversary arbitrarily perturbing this critical message. On the contrary, AME avoids overly depending on any single message by taking the ensemble of different message subsets.
>
> **2. AME is flexible and can be extended to environments with less information redundancy.**
>
> In environments without much information redundancy, the original AME algorithm may be too conservative, as some important yet sparse information may be overlooked. The problem is that the message-ablation policy $\hat{\pi}$ may not achieve high natural performance.  However, we can still extend AME to these environments without sacrificing much natural performance in the following ways.
>
> **(1) Increase $k$ to obtain more information in the ablated message samples.** This is the most straightforward solution based on the original AME, which has less dependency on information redundancy and can in general lead to higher natural performance.
>
> **(2) Increasing information redundancy when learning communication.** In tasks where information redundancy is insufficient to achieve robustness, one can also customize the communication protocol/policy such that a certain level of redundancy exists (e.g., increase the communication frequency). Note that both the message-ablation policy and the communication policy can be monitored and tuned in training time, so that the user can derive how robust the test-time policy is, based on Corollary C.1 or Theorem C.3. Our analysis on hyperparameters quantifies the needed information redundancy, making this solution easy to use in practice.
>
> **(3) Mixing AME and vanilla policies.** In environments and communication systems where information is intrinsically sparse (consider using multiple drones to find a single object in an area), one can equip a proportion of agents with AME while the other with the vanilla policy. For example, if a drone tells others it sees the object, vanilla drones would directly go to the reported location while the AME drones may choose not to go until they get more trustworthy consensus. Compared to all-vanilla policies which take all drones to the reported location and can be largely misled by adversaries, such a mixed strategy can better balance the natural performance and robustness.
>
> In addition, we would like to emphasize that MARL environments can vary a lot in terms of their structures and properties, and thus designing a universal approach that works perfectly for all environments is hard. However, our AME is based on a relatively simple intuition that can easily extend to many different settings as discussed above, which is also an important contribution of this work. We hope that our work can inspire more follow-up studies and methods to improve the robustness of multi-agent communication, an important yet underexplored topic.
>
> ---
>
> > Q7: The citation style after MARL-MNIST and Traffic Junction is inconsistent (page 8).
>
> Thank you for pointing it out. We have fixed the citations in the revised paper.
>
>
> ---
>
> We hope the above answers have addressed the questions of the reviewer. We are happy to discuss further if there are any other questions. Thank you again for reviewing our paper!

---

> > ### Comment · Reviewer_iqJ1 · 2022-11-18
> > **Reply**
> >
> > Thank you for the presentation updates and the through discussion.

---

> ### Author Response · Authors · 2022-11-16
> **Response to Reviewer iqJ1 - Part I**
>
> We are grateful that Reviewer iqJ1 precisely summarized our paper and provided very insightful feedback. Below we address the questions and concerns raised by Reviewer iqJ1 in detail.
>
> ---
>
> > Q1: The paper does not discuss the robustness certificate in depth (although the appendix does). Generally, I feel, the paper might benefit from including more from appendices B and C in the main paper (at the cost of shortening some other discussion).
>
> Thank you for the suggestion. We have moved more algorithmic details and theoretical interpretations from the appendix to the main paper. Due to the space limit, some details are still in the appendix, but we made sure that all important points are made in the main paper. Please let us know if there are more suggestions on the writing and paper organization. Thank you!
>
> ---
>
> > Q2: While the evaluation is thorough, Section 6 was hard to follow (as it required a lot of going back and forth between multiple paragraphs and Figure 3).
>
> Thank you for the suggestion! We have modified the layout to make this section easier to read. We also added a summarizing sentence in the figure caption such that the reader can get the main message from the figure itself.
>
> ---
>
> > Q3: I think the presentation could be improved by including at least Algorithm 2 in the main paper.
>
> Thank you for the comment. We have moved both Algorithm 1 and Algorithm 2 from the Appendix to the main paper.
>
> ---
>
> > Q4: To recover the exact experiments, I encourage the authors to provide their code to the eventual readers.
>
> We have provided the code for FoodCollector experiments in the supplementary material. We will make the code of all environments and detailed instructions publicly available when the paper is de-anonymized.
>
> ---
>
> > Q5: I assume that in the evaluation all agents employ AME. Is this the case? It would be potentially interesting to see settings where an AME agent interacts with Vanilla ones. Can you comment on this setting?
>
> Yes, all agents employ AME in our experiments since we aim to protect all agents from attacks.
>
> It is indeed a very interesting point to let an AME agent interact with Vanilla ones. Thank you for the suggestion! Below we provide detailed analysis and additional empirical results for this setting.
>
> **1. Analysis**
>
> As the paper has discussed in both theory and experiment, there is a trade-off between robustness and natural performance. If all agents adopt AME, then everyone can enjoy strong robustness, while the natural performance may be compromised. If some agents adopt AME while others execute vanilla policies, the averaged natural performance can be higher, and the AME agents can tolerate adversarial attacks; but the vanilla agents may fail under adversarial attacks. Therefore, this mixture of AME and Vanilla lies in between purely AME and purely Vanilla policies, in terms of both natural performance and robustness.
>
> This setting can be useful in practice. For example, if we know which agents or communication channels are more vulnerable to attacks, we can apply AME to these agents while keeping other agents vanilla. Also, in environments with little information redundancy, it is a good idea to deploy AME in a proportion of agents instead of all agents, to keep a certain level of communication efficiency. More discussion on information redundancy in the answer to Q6.
>
>
> **2. Empirical Results**
>
> Although this setting is intuitive in theory, we still conducted additional experiments to verify the above analysis. We take the Traffic Junction environment as an example, and use AME in only one agent. We name this variant as one-AME, while the original method all-AME.
>
> First, the average score of the all-AME agents is -3.58, while the one-AME variant gets -3.08, which shows that the average natural performance is increased by the application of one-AME. On the other hand, if there exist attacks (we use heuristic attacks here), the agent who adopts AME gets -3.54 when it is the victim; however, if an unprotected vanilla agent becomes the victim of attack, its reward significantly drops to -4.68, which verifies our analysis above.

---

### Official Review · Reviewer_drA6 · 2022-10-27

**Confidence:** 2
**Clarity, Quality, Novelty And Reproducibility:** 1)Author motivates the subject from a…
**Correctness:** 4
**Technical Novelty And Significance:** 3
**Empirical Novelty And Significance:** 3
**Recommendation:** 8

**Strength And Weaknesses:**

1)Author motivates the subject from a theoretical and practical point of view.
2)In the introduction, the author reviewed the previous works carefully.
3)the author successfully addressed the issue and recommended the algorithms.
 4)The authors successfully provide simulation results for their algorithm.
5)The proof is mathematically correct.
6) the paper is well-written.

**Summary Of The Paper:**

This paper studies communication attack in multi-agent reinforcement learning. They provide a defense by constructing a message-ensemble policy that aggregates multiple message sets. THey showed the effectiveness of their algorithm theoretically and practically.

**Summary Of The Review:**

This paper studies communication attack in multi-agent reinforcement learning. They provide a defense by constructing a message-ensemble policy that aggregates multiple message sets. THey showed the effectiveness of their algorithm theoretically and practically.

---

> ### Author Response · Authors · 2022-11-16
> **Response to Reviewer drA6**
>
> We greatly appreciate the valuable feedback of Reviewer drA6, and we are encouraged that Reviewer drA6 recognized the correctness of our theoretical results and the effectiveness of our algorithm.
>
> We have revised the paper according to the suggestions of reviewers. Please let us know if there are any questions. Thank you again for reviewing our paper!

---

### Official Review · Reviewer_BxRb · 2022-10-30

**Confidence:** 4
**Clarity, Quality, Novelty And Reproducibility:** The manuscript is overall in very goo…
**Correctness:** 4
**Technical Novelty And Significance:** 2
**Empirical Novelty And Significance:** 4
**Recommendation:** 6

**Strength And Weaknesses:**

Pro:

1. The manuscript investigates an interesting problem and the problem is indeed relevant in a variety of applications.

2. The authors have spent an effort to illustrate the proposed method from multiple aspects, including some theoretical derivation, some simulations, and the intuition underlying the method.

3. Experiments are conducted in multiple tasks.

Cons:

1. The guarantee of the "certifiable robustness" is very general and is not strong enough. I'm wondering if certain bounds on robustness and utility could be proved in concrete examples.

2. The setting that an agent receives $n$ messages is not realistic in many tasks, like SMAC. It is unlikely that all messages from $n$ agents are (equally) useful so in practice an attack on an important message could do more damage than the paper guaranteed.

**Summary Of The Paper:**

This manuscript introduces a new setting in multi-agent reinforcement learning with communication, where the communication could be attacked in several ways. The message could be alternated, the agents who send the messages could be hacked, and such attacks could happen in multiple places and could be adaptive to the receiver agent. To address this, the manuscript proposes a pretty much *general* solution, which is based on a message aggregation strategy. It basically guarantees that the receiver receives multiple messages and works on the ensemble of them, in a way that if some messages are attacked in some way, the aggregation still gives the close-to-true value. The manuscript claims that if the number of attacked messages is less than half then the action is to some extent "safe". The manuscript also did some simulations.

The method is a consensus-based aggregator, which trains a base policy that uses $k$ out of $n$ messages from the sender agents. At test time $k$ messages are sampled from all messages to compute the output and this process is repeated for multiple times. The median of the outputs is used as the final output. This method is quite intuitive.

**Summary Of The Review:**

A manuscript that investigates a novel and relevant setting. The results are decent yet limited to certain settings. I would say, this manuscript is borderline.

---

> ### Author Response · Authors · 2022-11-16
> **Part II: Additional Discussions about Message Importance**
>
> (Continuing A2 from Part I) Regarding the question of message importance, we provide additional explanation and discussion about the relationship between AME defense and message usage.
>
> **3. Additional Discussion: Trade-off between communication efficiency and policy robustness.**
> The reviewer makes an insightful point that different messages may have different importance to the agent. It is possible that sometimes an agent only needs to look at one message to make a good decision. However, if the decision is based on a single message, there is no hope to perform well if there is a strong adversary arbitrarily perturbing this critical message. This is exactly what AME is designed to avoid. By taking the ensemble of different message subsets, AME would not overly rely on any single message. (In training, AME trains the message-ablation policy $\hat{\pi}$ to maximize reward with ablated message samples, such that the agent does not overly rely on any single message.) Therefore, there is a trade-off between the communication efficiency and the policy robustness. Our AME, as a defense method, emphasizes more on the robustness side, at a price of lower communication efficiency (higher information redundancy).
>
> However, in some environments where valid communication is sparse, it could be too conservative if we make decisions only via consensus from subsets of messages. We admit that it is the limitation of our current AME algorithm. But it is not because the robust guarantee does not hold. It is because the message-ablation policy $\hat{\pi}$ may not obtain high natural performance when it drops the important yet sparse message. We note that in this case, it is still possible to adopt AME without sacrificing much natural performance in the following ways.
>
> **(1) Increasing information redundancy when learning communication.** In tasks where information redundancy is insufficient to achieve robustness, one can also customize the communication protocol/policy such that a certain level of redundancy exists (e.g. increase the communication frequency). Note that both the message-ablation policy and the communication policy can be monitored and tuned in training time, so that the user can derive how robust the test-time policy is, based on Corollary C.1 or Theorem C.3. Our analysis on hyperparameters quantifies the needed information redundancy, making this solution easy to use in practice.
>
> **(2) Mixing AME and vanilla policies.** In environments and communication systems where information is intrinsically sparse (consider using multiple drones to find a single object in an area), one can equip a proportion of agents with AME while the other with the vanilla policy. For example, if a drone tells others it sees the object, vanilla drones would directly go to the reported location while the AME drones may choose not to go until they get more trustworthy consensus. Compared to all-vanilla policies which take all drones to the reported location and can be largely misled by adversaries, such a mixed strategy can better balance the natural performance and robustness.
>
> In addition, we would like to emphasize that MARL environments can vary a lot in terms of their structures and properties, and thus designing a universal approach that works perfectly for all environments is hard. However, our AME is based on a relatively simple intuition that can easily extend to many different settings as discussed above, which is also an important contribution of this work. We hope that our work can inspire more follow-up studies and methods to improve the robustness of multi-agent communication, an important yet underexplored topic.
>
>
> ---
>
>
>
> Thank you again for reviewing our paper. We hope our clarifications and explanations have addressed your concerns and questions. If so, we would appreciate it a lot if you could consider raising the score. Please let us know if there are any other questions!
>
>
> ---
> Refs.
>
> [1]Rundong Wang, Xu He, Runsheng Yu, Wei qiu, Bo An, and Zinovi Rabinovich. Learning Efficient Multi-agent Communication: An Information Bottleneck Approach. In International Conference on Learning Representations, 2020.
>
> [2]Sai Qian Zhang, Jieyu Lin, Qi Zhang. Succinct and Robust Multi-Agent Communication With Temporal Message Control. In Advances in Neural Information Processing Systems, 2022.
>
> [3]Sai Qian Zhang, Qi Zhang, Jieyu Lin. Efficient communication in multi-agent reinforcement learning via variance based control[J]. In Advances in Neural Information Processing Systems, 2019.
>
> [4]Daewoo Kim, Sangwoo Moon, David Hostallero, Wan Ju Kang, Taeyoung Lee, Kyunghwan Son, and Yung Yi. Learning to schedule communication in multi-agent reinforcement learning. In 7th International Conference on Learning Representations, ICLR 2019.

---

> ### Author Response · Authors · 2022-11-16
> **Part I: Clarification on Our Guarantees and Methods**
>
> We greatly appreciate Reviewer BxRb for the valuable feedback and suggestions. We are encouraged that Reviewer BxRb finds the setting interesting and novel, and our paper in very good clarity and quality. We address the concerns raised by the reviewer below.
>
> > Q1: The guarantee of the "certifiable robustness" is very general and is not strong enough. I'm wondering if certain bounds on robustness and utility could be proved in concrete examples.
>
> A1: Thanks for the insightful question. We address this point from the following 3 aspects.
>
> 1. We indeed have analyzed how the robustness bound depends on the properties of the environment and the base message-ablation policy in Appendix C.2 in the case of a continuous action space. Our analysis shows that, if the **environment is smooth** (i.e., the agent does not immediately fail due to a single-step mistake), and the **benign agents achieve consensus** (i.e., the range of benign actions is small), then the attacked reward has a higher lower bound. Note that such "consensus" can actually be achieved by proper training, as discussed in A2 of this response below. In this case, our robustness guarantees are strong and tight. We have modified the main paper to make this point more clear.
>
> 2. Our purpose is to provide an easy-to-use defense mechanism and **general** robustness guarantees, such that the **theory works for any environment** instead of only in specific environments.
>
> 3. This paper provides the **first** certifiably robust algorithm against communication attacks. Please note that the studied threat model is a challenging setting, where the adversary can arbitrarily perturb one or more messages received by the victim agent, and the agent does not know the true information due to partial observability. Therefore, providing general and strong guarantees in this case is intrinsically hard, especially when the environment has high stochasticity. We believe that our algorithm, our robustness guarantees, and the analysis on hyperparameter selection, can benefit the community and inspire follow-up works on certifiably robust MARL communication in various settings.
>
> > Q2: The setting that an agent receives n messages is not realistic in many tasks, like SMAC. It is unlikely that all messages from agents are (equally) useful so in practice an attack on an important message could do more damage than the paper guaranteed.
>
> A2: Thank you for the question. There might be some misunderstandings on the assumption and the methodology. We would like to clarify the following points of our proposed AME defense algorithm.
>
> **1. Clarification: AME does not restrict that an agent receives n messages.**
> As mentioned in Sec. 3, we discuss the case where an agent receives $N-1$ messages just for illustration briefness. *Our algorithm and guarantees still hold if an agent receives messages from only a subset of other agents.* For example, if an agent receives $M<N-1$ messages at a step, our proposed AME can be operated over the $M$ messages and tolerate up to $C<M/2$ adversarial messages.
> In practice, AME can be used as long as the communication is not too sparse (an agent receives multiple messages), which is common in MARL environments. In SMAC, communication can be frequent and dense among certain agents in certain periods. For example, [1] observes that most of the communication appears in the beginning of the episode, due to the fact that the agents need to talk in order to arrange their positions. In such cases, communication attacks can potentially harm the agents, and thus robustness is important. In addition, [2] evaluates the communication overhead (i.e., proportion of agent pairs which conduct communication within a single timestep) of TMC[2], VBC[3] and SchedNet[4] as around 0.2, 0.3 and 0.5, which suggests that existing communication on SMAC is relatively dense, and information redundancy exists. Therefore, AME can be applied to improve the robustness of agents.
>
>
> **2. Clarification: AME is independent of how important each message is, and it stays robust even when important messages are attacked.**
> We respectively disagree with the argument that "an attack on an important message could do more damage than the paper guaranteed." Please note that our AME randomly drops out messages and takes the consensus of different ablated message sets, no matter how much weight the base ablation policy $\hat{\pi}$ puts on each message. As a result, even if an important message is perturbed, our robustness guarantee still holds (i.e.,test-time performance is similar to clean performance).
> In fact, if the agent makes decisions purely based on a single message, it is very risky if this message gets perturbed. In contrast, our AME encourages the agent to make decisions based on the ensemble of multiple information sources instead of putting too much weight on one single message. More insights on the rationale and extension of this design are in Bullet 3 (see next response).

---

> ### Author Response · Authors · 2022-11-28
> **Does our response address your concerns?**
>
> Dear Reviewer BxRb,
>
> As the first stage of the review discussion has ended, we would like to kindly ask you to review our response as well as the revised paper and consider making adjustments to the scores. Please let us know if there are any other questions. We would appreciate the opportunity to engage further if needed.
>
> Best regards,
>
> Paper3412 Authors

---

> > ### Comment · Reviewer_BxRb · 2022-12-01
> > **Response to the authors**
> >
> > My concerns are partially addressed. Some concerns remain on if the results are strong enough. To me, the setting seems to be limited, and a bit artificial. Nevertheless, as the authors pointed out this is the first study on the problem, and the paper is obviously well-written and is fun to read. I guess it would be nice if the community could find this paper at the conference.

---

> > > ### Author Response · Authors · 2022-12-04
> > > **Thank you for the response**
> > >
> > > Thank you for acknowledging the contributions of this work and raising the score! We are encouraged that you find our paper helpful for the community. Establishing a certifiable defense against strong communication attacks is an important and difficult problem, but it is not well-explored by existing literature yet. Therefore, we believe that this work can inspire more follow-up studies on certifiable robustness of MARL communication in a variety of settings. We will also consider extending the current method in our future work.
> > >
> > > Again, we greatly appreciate your valuable feedback and constructive suggestions. We are more than happy to discuss further if there are any other questions.
> > >
> > > Thanks,
> > >
> > > Paper3412 Authors

---

### Author Response · Authors · 2022-11-18
**General Response to All Reviewers**

We greatly appreciate the valuable and insightful feedback from all reviewers. We are particularly encouraged that reviewers find our problem relevant (BxRb, iqJ1), the theoretical results correct and sound (drA6, iqJ1), the empirical evaluation thorough (iqJ1), and the writing clear (BxRb, drA6, iqJ1).

We have replied to each reviewer with detailed answers and explanations to all their questions and suggestions. We have also updated the paper based on the feedback. The major changes of the paper are highlighted in blue color, and are summarized below.

- We added Algorithm 1 and Algorithm 2 to the main paper (Section 4.1) to make the algorithmic design clearer to the readers.
- We provided more theoretical analysis in Section 4.2 to better interpret the derived certificates.
- We illustrated the theoretical guarantees of the partial-sample variant of AME in Section 4.3.
- We adjusted the writing of Section 6 and provided more explanations in figure captions, to better present the experimental results.
- We added some discussion on future work about controlling information redundancy in Section 7.

We hope our responses and updates have addressed all concerns/questions that the reviewers have. Please let us know if there are any further questions!

---

### Public Comment · ~Ezgi_Korkmaz2 · 2023-02-15
**Acknowledgement of Recent Studies**

It would be reasonable for this paper to refer to recent studies [1,2,3] on adversarial deep reinforcement learning. When certified adversarial training methods are explicitly referred to, it should also be mentioned that the recent studies have already shown that certified adversarial training techniques are vulnerable to many different sets of attacks from perturbations that can transfer [2] to natural directions [1]. This study could acknowledge and refer to these studies.

[1] Adversarial Robust Deep Reinforcement Learning Requires Redefining Robustness. AAAI Conference on Artificial Intelligence, 2023.

[2] Deep Reinforcement Learning Policies Learn Shared Adversarial Features Across MDPs. AAAI Conference on Artificial Intelligence, 2022.

[3] Investigating Vulnerabilities of Deep Neural Policies. Conference on Uncertainty in Artificial Intelligence (UAI), Proceedings of Machine Learning Research (PMLR), 2021.

---

> ### Author Response · Authors · 2023-02-15
> **Thank you for the advice.**
>
> Thank you for your suggestions. They are all good studies in the area of adversarial RL. We will cite these papers in our camera ready version.

---

### Decision · Program_Chairs · 2023-01-20

**Decision:**

Accept: poster

**Justification For Why Not Higher Score:**

This seems to be not a very technically deep paper. It barely meets the publication bar.

**Justification For Why Not Lower Score:**

All reviewers lean towards acceptance.

**Metareview: Summary, Strengths And Weaknesses:**

This paper presents a certified defense algorithm against communication corruption on multi-agent RL. The strength of the paper includes its clean presentation and solid theoretical and empirical results. The weakness of the paper is that the studied setting is too general to allow for deep theoretical studies and that it doesn't allow any stochastic communications. I would recommend acceptance nonetheless.

**Note From Pc:**

if the above contains the word "oral" or "spotlight" please see: "oral" presentation means -> notable-top-5% and "spotlight" means -> notable-top-25%. As stated in our emails, we are disassociating presentation type from AC recommendations